# An adaptive biomolecular condensation response is conserved across environmentally divergent species

Samantha Keyport Kik [1], Dana Christopher[2], Hendrik Glauninger [3,4], Caitlin Wong Hickernell[2], Jared A. M. Bard [2], Kyle M. Lin [3,4], Allison H. Squires [5,6], Michael Ford[7], Tobin R. Sosnick [2,6] & D. Allan Drummond [2,6,8] ✉

Cells must sense and respond to sudden maladaptive environmental changes—stresses—to survive and thrive. Across eukaryotes, stresses such as heat shock trigger conserved responses: growth arrest, a specific transcriptional response, and biomolecular condensation of protein and mRNA into structures known as stress granules under severe stress. The composition, formation mechanism, adaptive significance, and even evolutionary conservation of these condensed structures remain enigmatic. Here we provide a remarkable view into stress-triggered condensation, its evolutionary conservation and tuning, and its integration into other well-studied aspects of the stress response. Using three morphologically near-identical budding yeast species adapted to different thermal environments and diverged by up to 100 million years, we show that proteome-scale biomolecular condensation is tuned to species-specific thermal niches, closely tracking corresponding growth and transcriptional responses. In each species, poly(A)-binding protein—a core marker of stress granules—condenses in isolation at species-specific temperatures, with conserved molecular features and conformational changes modulating condensation. From the ecological to the molecular scale, our results reveal previously unappreciated levels of evolutionary selection in the eukaryotic stress response, while establishing a rich, tractable system for further inquiry.

In response to a rapid increase in temperature—heat shock—eukaryotic cells respond by transcriptionally inducing a conserved set of genes encoding molecular chaperones[1–3], repressing cell growth and translation[4], and accumulating protein and mRNA molecules in large clusters called stress granules[4–7]. Until recently, these actions have been conceived of as a response to protein damage, denaturation, and aggregation caused by heat, with chaperones acting to restore protein homeostasis[8–10].

However, substantial evidence now supports an alternative model describing the events following heat shock: physiological changes in temperature are directly sensed by specific proteins, triggering their biomolecular condensation without large-scale damage or global

[1]Committee on Genetics, Genomics, and Systems Biology, The University of Chicago, Chicago, IL, USA. [2]Department of Biochemistry and Molecular Biology, The University of Chicago, Chicago, IL, USA. [3]Graduate Program in Biophysical Sciences, The University of Chicago, Chicago, IL, USA. [4]Interdisciplinary Scientist Training Program, The University of Chicago, Chicago, IL, USA. [5]Pritzker School of Molecular Engineering, University of Chicago, Chicago, IL, USA. [6]Institute for Biophysical Dynamics, University of Chicago, Chicago, IL, USA. [7]MS Bioworks, Ann Arbor, MI, USA. [8]Department of Genetic Medicine, The University of Chicago, Chicago, IL, USA. ✉e-mail: dadrummond@uchicago.edu

denaturation[7]; condensation is adaptive rather than deleterious, even affecting several of the translational changes observed[11,12]; chaperones act as regulators of the condensation response[13–15]; and temperature itself often acts as a physiological signal carrying adaptive information[16]. Remarkably, although adaptive biomolecular condensation was recognized last among these phenomena, it plays a central role in each aspect of the response, leading to the prediction that condensation should be both conserved and intimately coordinated with other aspects of the response across species—a prediction which motivates the present study.

To more clearly see the proposed interrelationships between condensation and more well-established aspects of the cellular response to heat shock, we consider transcriptional upregulation of specific genes, a defining feature of the response since its discovery[17,18]. The core eukaryotic heat shock response (HSR) is regulated by heat shock factor 1 (Hsf1), which induces transcription of several heat shock proteins, including the molecular chaperones of the Hsp70 family. Under physiological growth conditions in *S. cerevisiae*, one or more Hsp70 proteins repressively bind the transcription factor Hsf1[13,14]. New and abundant Hsp70 substrates emerging during stress are thought to titrate Hsp70 away from Hsf1, relieving inhibition and unleashing the HSR. Stress-induced protein misfolding was long thought to generate these substrates, and significant evidence indicates that misfolded proteins are sufficient to induce Hsf1 at non-shock temperatures[19], yet misfolding of mature endogenous proteins under physiological heat-shock conditions has remained surprisingly difficult to establish. Meanwhile, evidence has accumulated that nascent polypeptides and newly synthesized proteins, whose folding and assembly may be more easily perturbed during stress[20–23], may serve as HSR inducers[16,24]. Nevertheless, the HSR can be robustly induced even when translation is inhibited, indicating that nascent/new species cannot be the sole HSR trigger[16,25]. Moreover, physiological biomolecular condensation in response to thermal stress has repeatedly been shown to be an autonomous property of individual proteins[7,11,12,26], and such condensates recruit Hsp70 both in vivo and in vitro[4,15], indicating that they may more broadly serve as physiological thermal sensors. Finally, many of the proteins which condense in response to stress are translation initiation, elongation, or ribosome biogenesis factors, whose condensation—and likely inactivation—accompanies suppression of the associated processes[4,7,27]. Together, these observations open the possibility that condensation may serve as a primary sensor, coordinator, and executor of the transcriptional and translational stress responses.

If condensation acts in this central organizing capacity then, as noted above, it should be conserved across related species and coordinated with the other aspects of their stress responses. For heat shock, the obvious candidate species are those adapted to different thermal niches: thermophiles, mesophiles, and cryophiles. Tuning of the transcriptional HSR to suit the environmental temperature profile and organism lifestyle is well-established[28–31]. That is, similar transcriptional programs are induced at different temperatures by organisms occupying different thermal niches.

Due to evidence suggesting that condensation may trigger the transcriptional heat shock response as described above, an obvious hypothesis is that the magnitude of condensation and of the transcriptional heat shock response will correlate across thermally adapted species. The thermal stability against misfolding of thermophilic proteins relative to their mesophilic and cryophilic counterparts has been repeatedly exploited, perhaps most famously in the polymerase chain reaction (PCR)[32]. Similarly, thermally triggered biomolecular condensation of orthologs of the RNA helicase Ded1 occurs at increasing temperatures for orthologs from a cryophilic, mesophilic, and thermophilic fungus respectively[12]. Whether this condensation influences fitness, and the nature of the relationship between species heat shock response temperatures and protein condensation temperatures, were not assessed, yet the imprint of thermal niche seems evident. Tuning of protein stability and condensation temperatures appear to track ecological adaptations.

How much room for tuning exists within physical constraints under which life evolves? Evolution appears to have strong constraints on its ability to craft extremophiles, with potentially important consequences for understanding stress, biophysical constraints, and the limits of organismal growth. No known eukaryote grows above 62 °C[33], and fungi are the only eukaryotes which grow above 45 °C[34], in contrast to the many bacteria and archaea capable of growing at substantially higher temperatures, even exceeding 100 °C at sufficient pressures to prevent the boiling of water. What causes this apparent thermal cap on eukaryotic growth is unknown[35]. Whatever the resolution to this great mystery, we cannot ignore the possibility of strong biophysical constraints on eukaryotic life beyond mere niche preference. Nevertheless, within such constraints, evolution enjoys substantial freedom to conserve cellular operations while tuning their execution to suit ecological opportunities.

Many related questions follow from these observations. How do growth, transcription, and condensation responses change together across organisms adapted to different thermal niches? To what extent has evolution conserved and tuned the response to temperature—particularly in the condensation response, for which only data for a handful of proteins exists? How similar are the molecules which form temperature-triggered condensates across species? How, if at all, are phenotypes conserved across thermally divergent species when condensation is perturbed? And how closely are the molecular details of condensation preserved among divergent homologs?

To answer these questions, we characterized the growth, transcription, and protein condensation of cryophilic, mesophilic, and thermotolerant budding yeast species across a range of temperatures and molecular scales. As expected, transcriptional heat shock responses track each species' thermal tolerance, which we characterize in detail. We show that at the proteome scale, protein condensation is tuned across species, such that the behavior of orthologous molecules is conserved but at niche-specific temperatures. We show that condensation is encoded in the amino-acid sequence of stress granule marker poly(A)-binding protein (Pab1) as shown by in vitro biophysical assays, and disrupting the condensation of Pab1 in vivo reduces fitness during temperature stress. Finally, at the molecular level, we employ hydrogen-deuterium exchange - mass spectrometry (HDX-MS) to compare structural differences in condensates relative to monomers across species, revealing new levels of conservation and divergence.

## Results

### Evolutionary divergence results in distinct thermal growth ranges

For comparison of conservation and divergence after heat shock under controlled conditions, we selected three budding yeast species (Fig. 1a) known to differ in their thermal profiles yet able to grow robustly on identical media containing D-glucose. Baker's yeast *Saccharomyces cerevisiae*, which grows optimally between 30 °C–34 °C, grows naturally on the surface of fruit and on oak bark, but halts its growth above 40 °C. Another related yeast, *Saccharomyces kudriavzevii*, also found on oak bark, grows optimally at lower temperatures between 24 °C–28 °C, and growth has been shown to cease above 32 °C[36]. We also used a pre-whole-genome duplication relative, *Kluyveromyces marxianus*, a respirative thermotolerant yeast isolated from sugarcane juice in Thailand, which can grow robustly at temperatures as high as 45 °C[37,38]. (We use the convention that reserves "thermophilic" for fungi which grow more rapidly at 45 °C than at 25 °C, while labeling others "thermotolerant"[39].) Consistent with previous studies, we hypothesized that each species' transcriptional changes would be tuned to the thermal environment to which they have adapted[30,31,40–43].

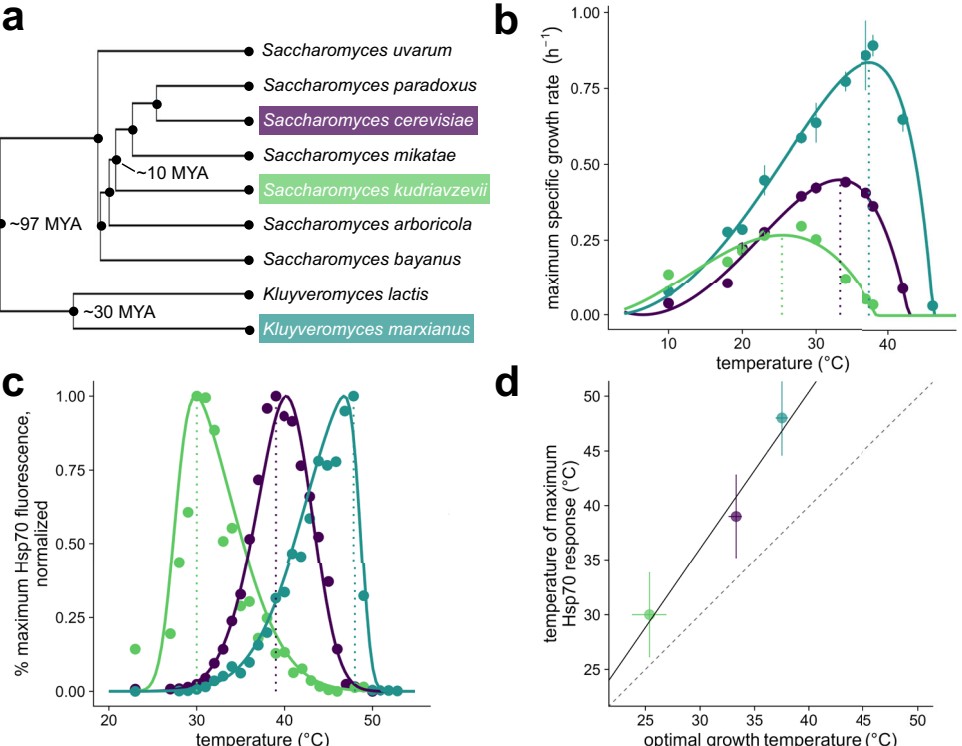

**Fig. 1 | Three fungal species grow optimally at different temperatures, and their transcriptional heat shock responses track their optimal growth temperatures. a** Phylogenetic tree of the Saccharomycetaceae family. The topology was obtained from Kumar et al. 2022. **b** Growth rate versus temperature, shown as mean +/− standard error, overlaid with a fit of the cardinal temperature model with inflection to experimental data obtained for strains *S. cerevisiae* BY4742 (purple), *S. kudriavzevii* FM1389 (green), and *K. marxianus* DMKU3-1042 (blue). Dotted lines show the temperature of the maximum growth rate for each species. At least two biologically independent cultures were measured for each temperature and species. **c** Hsp70 fluorescence after 20 minute heat shock at specified temperature

with three hours of recovery. Values are plotted as the percent of the maximum response. Each point is the average of at least 5000 cells, controlled for size and normalized to non-heat shocked cells. Dotted lines show the temperature of the maximum measured transcriptional response for each species. **d** Correlation of the temperatures at which the three species reach their highest maximum specific growth rate from **b** plotted against the temperature of the maximum Hsp70 response in **c**. Error bars represent the standard deviation of temperature for the fit of the skew-normal distribution (y-axis) or cardinal temperature model with inflection (x-axis). Solid line shows a linear fit, dashed line shows $y = x$. Source data are provided as a Source Data file.

To characterize growth behavior, we calculated maximum growth rates of log phase cultures at temperatures ranging from 10 to 46 °C for at least two biological replicates (Fig. 1b). *K. marxianus* shows a high maximum growth rate relative to these other species, consistent with other studies on this organism[37,44–46] and likely due to its respirative life history[38,47]. The estimated growth temperature optimum is different for each species, ranging from 25 °C–38 °C. Observations of growth in liquid culture also match longer-term growth on plates (Figure S1a). In contrast to previously published results, for *S. kudriavzevii*, we do observe growth above 32 °C, as high as 38 °C, although very slowly and specifically in complete medium liquid culture[48].

These growth studies yield clear species contrasts: at 37 °C, the standard heat-shock temperature commonly used in studies of mesophilic baker's yeast, the thermotolerant species has not yet reached its optimal growth temperature—in this sense, it is truly heat-loving rather than merely heat tolerant—while growth of the cryophilic species has all but ceased.

## Heat shock responses track temperature tolerance

How do these species' heat shock responses relate to the temperatures at which they proliferate? To monitor the heat shock response quantitatively, we constructed *S. cerevisiae* and *S. kudriavzevii* strains with endogenous heat-induced Hsp70 molecular chaperone *SSA4* tagged with the red fluorescent protein mCherry[49], as well as a *K. marxianus* strain expressing the pre-duplication ortholog of the *SSA3* promoter driving eGFP. We monitored red or green fluorescence using flow cytometry after a 20-minute temperature treatment followed by a

three-hour recovery at room temperature (Fig. 1c). Each population shows some variance at each temperature, with less variance observed at heat shock temperatures relative to non-heat shock temperatures for both species (Figure S2a). As expected, the temperature of maximum response is lowest for *S. kudriavzevii*, intermediate for *S. cerevisiae*, and highest for *K. marxianus*.

The data reveal a strong correlation between optimum growth temperature and maximum Hsp70 induction temperature across the three species (Fig. 1d). The difference between the maximum heat shock response temperature and the optimum growth temperature varies between 6 °C–8 °C, where growth rate has begun to decline in each species but has not reached complete growth arrest.

We next wanted to know if the observed Hsp70 response generalizes to the broader HSR, including Hsf1-regulated genes, and to characterize other gene expression changes across species. To measure the global transcriptomic behavior, we performed RNA-seq for each species with two biological replicates, comparing transcript levels at the optimum growth temperature to those at the maximum heat shock response temperature (Figure S3a). Overall, many of the transcriptomic changes are conserved at each species' respective heat shock temperatures (Fig. 2a, Figures S3b, c), with genes encoding orthologous ribosomal components, ribosome biogenesis factors, and translation factors downregulated in each species (Supplementary Files 1 and 2, Methods). Likewise, orthologs controlled by heat shock factor 1 (Hsf1) are strongly induced in all three species.

Environmental stress response (ESR) factors Msn2 and Msn4 (Msn2/4) are paralogous transcription factors in fungi which become

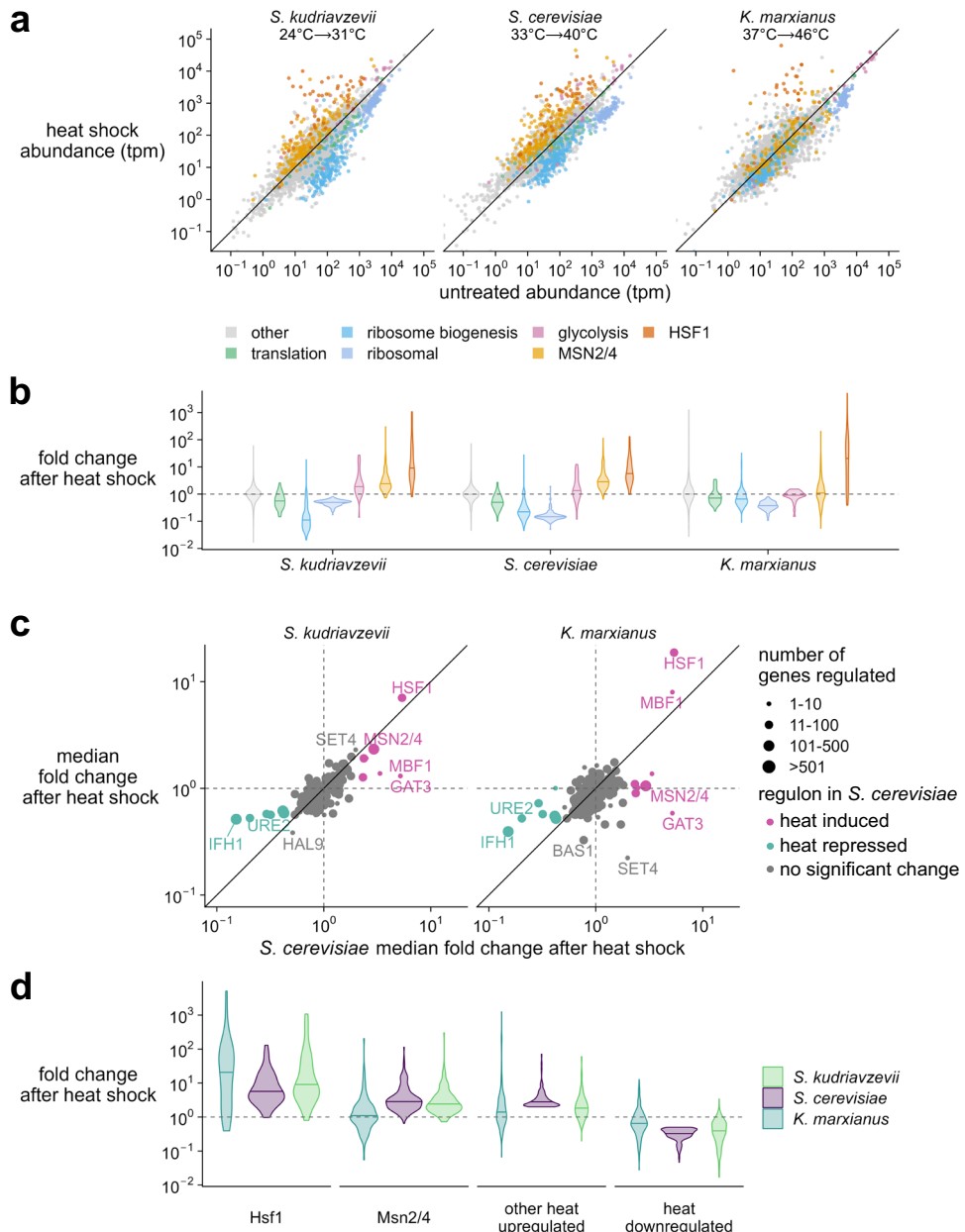

**Fig. 2 | Transcriptome changes upon heat shock reflect largely conserved responses to stress, except environmental stress response regulators Msn2/4.** **a** Transcript abundance (transcripts per million, tpm) in stressed versus unstressed populations of cells. *S. kudriavzevii* was grown at 24 °C and stressed at 31 °C; *S. cerevisiae* was grown at 33 °C and stressed at 40 °C; *K. marxianus* was grown at 37 °C and stressed at 46 °C. Colors correspond to gene type. The *y* = *x* line is shown in black. **b** Fold change distribution for groups of genes (colored by gene type) after stress in each species. **c** Behavior of genes corresponding to orthologous transcription factor regulators in *S. cerevisiae*. Data point size corresponds to number of genes controlled by the regulator, and points are colored according to genes that were observed as two-fold up or down in *S. cerevisiae*. The *y* = *x* line is shown in black. **d** Individual transcript fold change after heat shock colored by species. Panels correspond to orthologous genes controlled by Hsf1 or Msn2/4 from *S. cerevisiae* (left two panels) or orthologous genes that were observed as two-fold up or down in *S. cerevisiae* (right two panels). Source data are provided as a Source Data file.

activated during many stresses, including heat, osmotic, and oxidative stress[50–54]. *S. cerevisiae* and *S. kudriavzevii* possess both paralogs, but *K. marxianus* only possesses one, as it diverged before the fungal whole genome duplication. Msn2/4 bind to stress response element (STRE) promoter sequences and have been shown to regulate more than 200 genes in *S. cerevisiae*, including several that are also induced by Hsf1[2,3]. After a 20-minute heat stress, Msn2/4 induces hundreds of genes in *S. cerevisiae* and *S. kudriavzevii* (Fig. 2b).

Interestingly, full ESR upregulation behavior does not seem to be conserved in *K. marxianus* across the Msn2/4 regulon; instead, many orthologs under Msn2/4 control in *S. cerevisiae* are instead

downregulated during heat shock, resulting in a near-zero average response. *K. marxianus* diverged from the *S. cerevisiae* lineage about 100 million years ago, prior to the whole genome duplication. Previous work has described the stress response in *Lachancea kluyveri*, another pre-duplication relative, observing little overlap in the heat-induced ESR response with *S. cerevisiae*[47], and proposing that this may be due to differences in life history: respiratory yeast like *K. marxianus* do not require the same ESR responses as fermentative yeast like *S. cerevisiae*. We therefore looked more closely at the subset of annotated *S. cerevisiae* Msn2/4 orthologous targets showing altered transcriptomic behavior and found consistent results with *K. marxianus*, such that a

substantial proportion of downregulated Msn2/4 orthologous targets in *K. marxianus* were also downregulated in response to heat shock in *L. kluyveri* (Figure S2b, c), consistent with the respiratory hypothesis.

To more broadly identify regulons whose heat-shock behavior might differ between species, we grouped transcripts by their orthologous transcription factors (Supplementary File 3, Methods) from *S. cerevisiae* (Fig. 2c). Most of these regulons do not respond to temperature in *S. cerevisiae*, but most that do correspond with regulons that change in *S. kudriavzevii* (Spearman's $\rho = 0.72$). However, a lower correlation is observed between *K. marxianus* and *S. cerevisiae* (Spearman's $\rho = 0.48$), with the median of orthologous Msn2/4 regulated genes showing no net change in the thermotolerant species. With the exception of Msn2/4, orthologous genes that are 2-fold up or downregulated in *S. cerevisiae* seem to be largely conserved across species, including Hsf1 regulated genes (Fig. 2d), raising the possibility that the upstream sensors of temperature carry much of the niche-specific tuning which is then transmitted to transcription factors.

In the case of Hsf1, these upstream sensors may include thermally triggered biomolecular condensates which titrate Hsf1-repressive Hsp70 to activate the Hsf1 portion of the HSR. Moreover, Hsf1 targets include many molecular chaperones, including Hsp70, known to regulate the dispersal of biomolecular condensates[4,15,55]. We therefore sought to measure condensation in these three species across their heat shock temperatures, asking to what extent condensation is conserved and tuned.

## Heat-shock-triggered protein condensation is conserved across species

To survey condensation behavior across species, we performed biochemical fractionation and LC-MS/MS before and after 8-minute, species-specific temperature treatments much as in our previous study (modifications detailed in Methods) in *S. cerevisiae*[7]. Previous work has indicated that after 8-minutes, there is no significant change in protein levels[7]. We estimate the proportion of a given protein in the supernatant (pSup) using a model which controls for experimental mixing error (Methods). Many purified proteins identified using this method have been shown to condense in vitro when exposed to physiological heat shock—including poly(A)-binding protein Pab1 and aminoacyl tRNA synthetase complex components Gus1, Arc1, and Mes1[7]; RNA helicase Ded1[12]; and poly(U)-binding protein Pub1[26]—with no known exceptions to date. Our MS method cannot distinguish which proteins cause condensation (drivers) rather than bind to such proteins (passengers). In the case of the Gus1/Arc1/Mes1 complex, Gus1 and Mes1 are drivers, whereas Arc1 is a passenger[7], yet all three condense. The question of which proteins form condensates is separate from the question of how condensation occurs. Given these findings, we interpret a change from high pSup to low pSup (i.e., high solubility to low solubility) during heat shock as indicating condensation. Nevertheless, it remains possible that certain proteins change pSup in this way due to shock-induced binding to membranes or other large structures.

Across the three species, we observe large-scale changes in protein condensation, primarily through decreases in pSup in heat shock relative to non-heat shock conditions (Fig. 3a, top). Among orthologous proteins, glycolytic enzymes, ribosomal components, and membrane proteins do not significantly shift their solubility, shown by the pSups falling near the 1:1 line between conditions (Fig. 3a, bottom). However, some orthologous proteins appear in the insoluble fraction after temperature stress, including a group of so-called superaggregator proteins identified previously[7] and so named because they lose solubility more rapidly during heat shock than many stress-granule components. Subsequent work has demonstrated that recombinant purified preparations of specific superaggregators, such as Ded1, undergo condensation in vitro in response to heat shock. We observe superaggregator orthologs almost entirely retain this extreme temperature sensitivity, but at temperatures that correspond to each

organism's thermal growth range. Superaggregators as a class display decreased solubility at 37 °C in *S. kudriavzevii*, for which this is a heat-shock temperature, yet retain high solubility at the same temperature in *K. marxianus*, for which 37 °C is a near-optimal growth temperature.

To quantify conservation of these responses, we calculated pSup correlations between conditions and among all three species. Under non-heat shock, basal conditions for *S. kudriavzevii* (23 °C) and *K. marxianus* (37 °C), we observe that protein classes largely display the same behavior ($R^2 = 0.93$, Fig. 3b, left). Likewise, after heat shock, the pSups of protein classes are again strongly correlated ($R^2 = 0.73$, Fig. 3b, middle). We observe consistently high correlations for all species when comparing within temperature conditions (e.g., heat shock vs. heat shock). Correlations drop significantly when comparing between treatment conditions (heat shock vs. basal), even within species (Fig. 3b, right); that is, the condensation status of *S. kudriavzevii* during heat shock is more similar to that of *K. marxianus* during heat shock than it is to that of *S. kudriavzevii* under basal conditions. We conclude that biomolecular condensation of specific proteins in response to an evolutionarily tuned heat shock is conserved.

Because we performed these experiments under species-specific basal and mild or more severe heat shock temperature conditions, we next aimed to evaluate temperature tuning. We focused on glycolytic proteins and superaggregators across three temperatures, because the former shows little temperature sensitivity and the latter the greatest. We observe a striking relationship between temperature and condensation of superaggregators: as temperature increases, pSup decreases (Fig. 3c). However, this occurs at different temperatures for each organism, and non-condensing glycolytic enzymes show consistent solubility irrespective of heat shock. Thus, condensation is not only conserved among evolutionarily distinct organisms, but it is tuned to their adapted temperature niche. Importantly, we conclude that biomolecular condensation of certain proteins occurs in response to heat shock among distinct species, and this phenomenon has evolved alongside environmental adaptation.

## Condensation of Pab1 is conserved, encoded in sequence, and environmentally tuned

Given previous results that some purified proteins from *S. cerevisiae* autonomously condense in response to heat shock, we wondered if this property was tuned in a purified protein from *S. kudriavzevii* and *K. marxianus*[11,12]. To test this, we purified poly(A)-binding protein (Pab1) from each species. The two Pab1 orthologs have varied differences in amino acid sequence divergence, with 98% identity between *S. cerevisiae* and *S. kudriavzevii* but only 69% between *S. cerevisiae* and *K. marxianus*.

As in previous work[11], we used dynamic light scattering (DLS) to monitor the apparent hydration radius ($R_h$) during a slow temperature ramp; baseline low-temperature $R_h$ values report on the monomer size, and sharp increases in $R_h$ mark the onset of condensation. Purified Pab1 from each species condensed at a temperature ($T_{condense}$, see Methods) correlated with both the optimal and maximum growth temperatures (Fig. 4a, b). DLS is highly sensitive to small condensates of only a few nanometers in size. We thus also looked for the appearance of microscopically visible condensates, which emerged at temperatures correlated with each species' thermal niche (Figure S4). Condensates in all species also had similar morphologies, branched chains of small clusters, as previously reported[11] (Figure S4). We conclude that Pab1's temperature sensitivity and tuning to a thermal niche are both largely encoded directly in its amino-acid sequence, paralleling results from the RNA helicase Ded1[12].

Previous work has shown that Pab1's P domain, a low-complexity region enriched for proline, modulates the temperature at which the *S. cerevisiae* protein forms condensates, but the domain is not required for condensation[11]. We wondered if this modulatory role was also conserved across species. Interestingly, the P domain sequences of *S.*

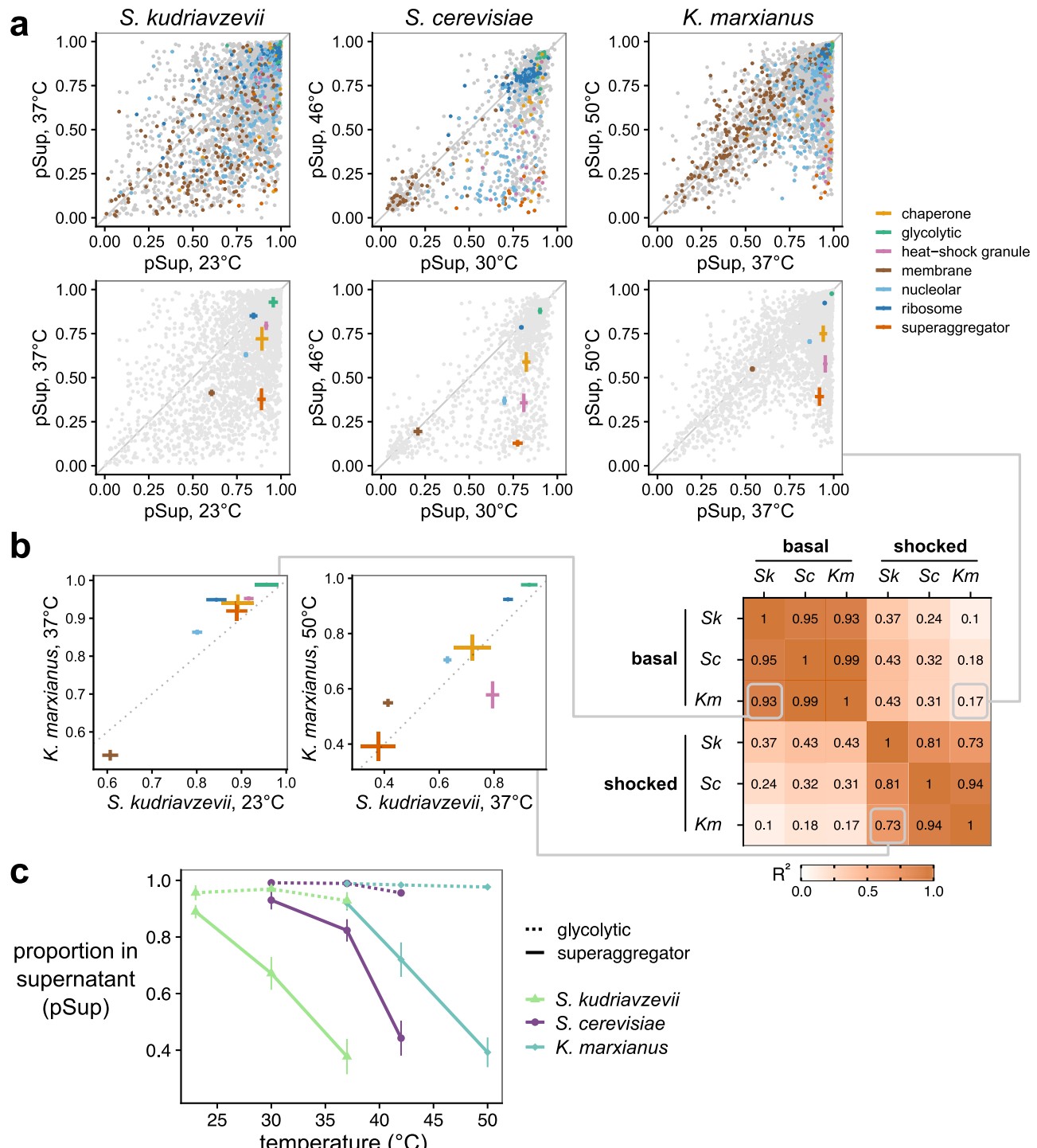

**Fig. 3 | Condensation is conserved across species and tuned to their respective thermal niches. a** Proportion in the supernatant (pSup) at basal temperature and after heat shock, at species-specific temperatures, across three fungal species. *S. cerevisiae* data are from Wallace et al. 2015. Top, all detected proteins, with important previously identified classes of proteins highlighted, inferred by orthology from *S. cerevisiae*. Bottom, summary statistics (mean +/− standard error) for the highlighted classes (for *S. kudriavzevii*, *S. cerevisiae*, and *K. marxianus* respectively, class sizes are: chaperone *n* = 18, 25, 22; glycolytic *n* = 11, 16, 18; heat-shock granule *n* = 19, 21, 20; membrane *n* = 193, 150, 221; nucleolar *n* = 236, 217, 255; ribosome *n* = 69, 103, 108; superaggregator *n* = 15, 17, 15). **b** Conservation of

condensation across classes of proteins is revealed by correlations between their condensation behaviors within and between species. Left, specific comparisons of basal and shocked condensation between *S. kudriavzevii* and *K. marxianus* at their respective optimal growth and shock temperatures; lines connect to their respective entries in the full correlation matrix (showing $R^2$ values), right. Crosses show mean +/− standard error. **c** Tuning of condensation is revealed by comparison of non-condensing glycolytic proteins and strongly condensing superaggregators across species. Error bars represent the mean +/− standard error. Source data are provided as a Source Data file.

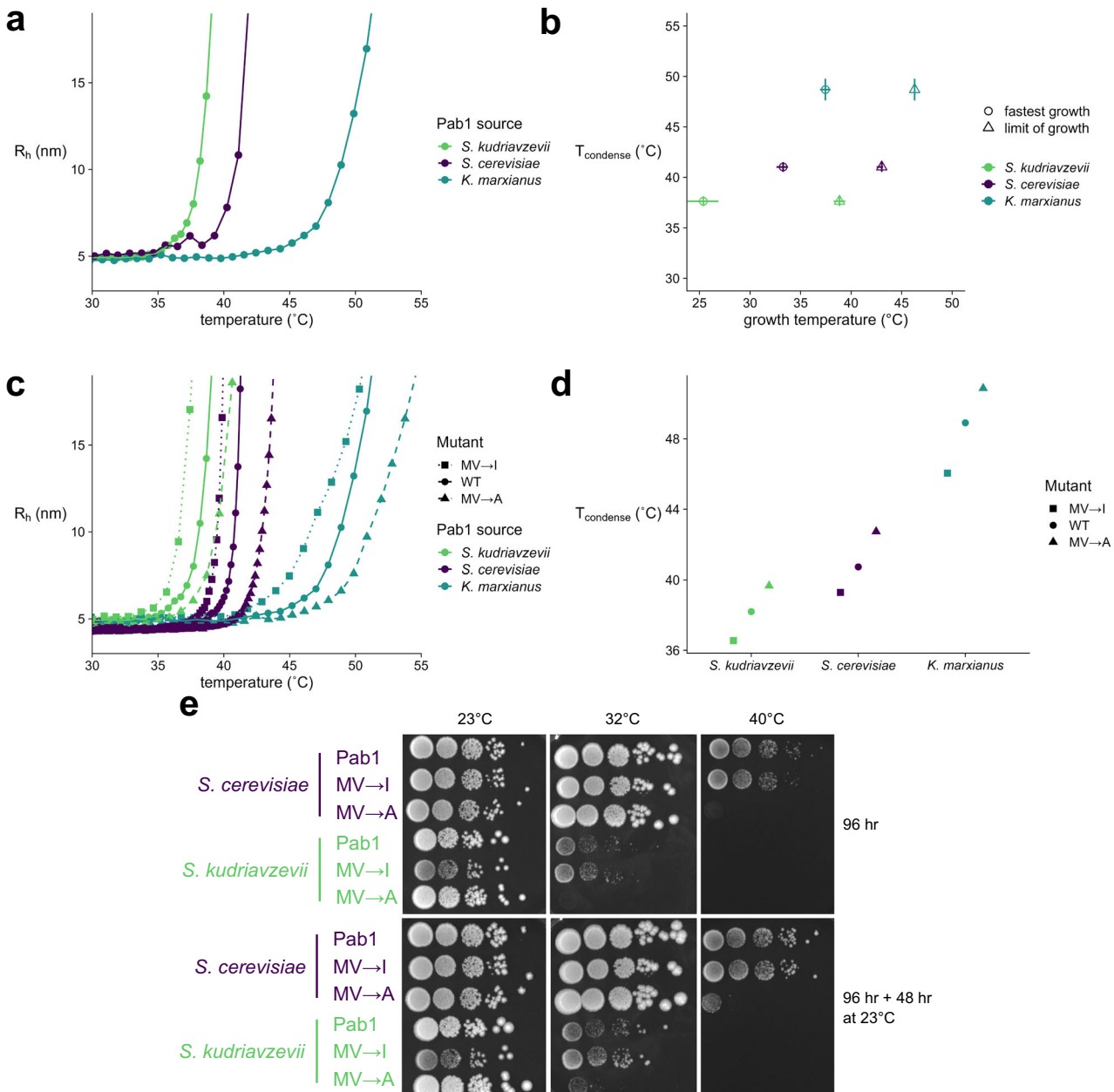

**Fig. 4 | A conserved yet environmentally tuned condensation response is adaptive across species with differing temperature optima for growth. a** DLS temperature ramp experiments of 15 μM Pab1 from each species, all at pH 6.4. **b** Correlation of $T_{condense}$ (°C) with optimum growth temperature (circles) or maximum growth temperature (triangles). Error bars for $T_{condense}$ (°C) represent the standard deviation of the mean of at least three replicates. Error bars for the optimum and maximum one standard deviation of the parameter estimates from the fit of the cardinal temperature model with inflection. Error bars for *S. cerevisiae* calculated from results are from n = 4, n = 5, and n = 6 for MV → I, WT, and MV → A, respectively. Error bars for *S. kudriavzevii* calculated from results are from n = 3, n = 4, and n = 3 for MV → I, WT, and MV → A, respectively. Error bars for *K. marxianus*

calculated from results are from n = 4, n = 5, and n = 4 for MV → I, WT, and MV → A, respectively. **c** DLS temperature ramp experiments of 15 μM Pab1 mutants from each species, all at pH 6.4. Twelve mutations were made for each mutant from each species. Curves for *S. cerevisiae* mutants and wildtype are from Riback & Katanski et al. 2017. **d** Comparison of $T_{condense}$ (°C) values calculated for each Pab1 protein. $T_{condense}$ (°C) was calculated for *S. cerevisiae* mutants and wildtype from the curves contained in Riback & Katanski et al. 2017. **e** Spot assays of *S. cerevisiae* and *S. kudriavzevii* strains containing mutations in the P domain. Plates were incubated at 23, 32, and 40 °C for four days (96 hours), and then shifted to 23 °C and grown for two days (48 hours). Columns are 10-fold dilutions. Source data are provided as a Source Data file.

*cerevisiae* and *S. kudriavzevii* orthologs are identical, yet their $T_{condense}$ values differ, which indicates that P-domain modulation of condensation is not responsible for differences between species.

A simple test for modulation, given our results in *S. cerevisiae*, would be to make the homologous mutations in the P domains of the other two species. This is straightforward in *S. kudriavzevii*, whose P domain is identical to *S. cerevisiae*, but nontrivial in *K. marxianus*, whose P domain is only 57% identical. We therefore adopted the same strategy

as in our previous study: mutating all instances of the weakly hydrophobic residues methionine and valine to isoleucine (MV → I), or to alanine (MV → A), making the domain more or less hydrophobic, respectively. We previously observed that $T_{condense}$ positively correlated with the hydrophobicity of each domain, raising the question of whether this specific biophysical change would have similar consequences.

Indeed, the MV → I mutants in both species showed lower-temperature onset of condensation relative to their species' wild

type, and the MV → A mutants showed higher-temperature onset, mirroring the *S. cerevisiae* results in both cases (Fig. 4c). We conclude that the P domain's modulation of condensation is conserved, and a major biophysical determinant of this modulation is also conserved, across all three species (Fig. 4d). This is true whether the domain itself is perfectly conserved or sharply divergent in sequence.

## Phenotypic consequences of altering condensation are conserved

Condensation of Pab1 at the appropriate temperature in *S. cerevisiae* is important for fitness of the organism during long-term heat stress[11]— that is, Pab1 condensation is adaptive. We wondered if this adaptive phenotype was conserved among species. To test this hypothesis, we replaced the Pab1 at its endogenous locus with the P domain mutants (MV → I or MV → A) in both *S. kudriavzevii* and *S. cerevisiae* (this genetic manipulation was not successful in *K. marxianus*). Consistent with our earlier results, we observe that *S. cerevisiae* strains grow at the same rates at species-specific basal temperatures of 23 °C and 32 °C, but at a heat shock temperature of 40 °C, the MV → A mutant has reduced fitness (Fig. 4e, Figure S2b). When returned to 23 °C for 48 hours, a substantial portion MV → A mutant cells remain viable, indicating that the lack of growth was not due to death of those cells but rather arrested growth. We observe consistent results in *S. kudriavzevii*, where at 23 °C, all strains grow equally, but at 32 °C—close to optimal growth for *S. cerevisiae*, but a strong heat shock for *S. kudriavzevii*—the strain bearing the less hydrophobic MV → A P domain mutant has reduced fitness, and resumes growth when the stress is relieved by returning to 23 °C for 48 hours.

Growth of *S. kudriavzevii* does not resume when cells are moved to 23 °C after stress at 40 °C (Fig. 4e). Why these cells cannot resume growth, and whether this behavior has any link to condensation, is unclear; *S. kudriavzevii* shows no growth at 40 °C (Fig. 1b), so after four days, the small number of initially deposited cells may simply have become inviable.

Together, these results suggest that the prevention of condensation in vivo decreases the ability of the organism to grow at temperatures above that of optimal growth, a phenotype which is consistent across species yet tuned to the evolved environment of the organism. Given the consistent effect of these modulatory mutations, both on condensation and on the resulting phenotype, we then wondered how the molecular structural dynamics of these orthologs might similarly be conserved and tuned.

## Structural features of monomers and condensates are conserved

To investigate the structural conservation of the monomers and condensates for the three purified Pab1 orthologs, we used hydrogen-deuterium exchange mass spectrometry (HDX-MS), building on our previous work on *S. cerevisiae*'s Pab1[56]. HDX-MS reports on a protein's hydrogen bond network involving amide protons across the protein by monitoring the rate of deuterium exchange, typically obtained at the peptide level using in-line proteolysis and mass spectrometry. Compared to solvent exposed amide protons, exchange occurs slower for sites involved in hydrogen bonds, and even more slowly— perhaps not at all on the timescale of the measurement—for sites in the most stable hydrogen bonds. In addition, and in contrast to many other structural techniques, HDX-MS can be used to study insoluble structures such as condensates, providing insight into their hydrogen bond network down to the peptide or even residue level.

To evaluate the extent of structural conservation for the three species, we compared the exchange behavior of monomeric Pab1 in each species, for which we were able to obtain excellent peptide coverage (Figure S5). We measured proportional deuterium uptake (%D) after 100 seconds of deuterium. The 100 sec time point is short enough that exchange does not measurably occur for the most stable

hydrogen bonds, but long enough that it can be observed in the least stable regions, which enables us to distinguish between stably folded and unstructured regions.

Indeed, we observed substantial variation in deuterium labeling between regions within each Pab1 ortholog. The highly structured RNA-recognition motifs (RRMs) exhibited the lowest levels of exchange while interdomain linkers, along with the intrinsically disordered proline-rich (P) domain, showed higher levels (Fig. 5a). Exchange levels at each site are highly correlated across orthologs (r > 0.91) for each pairwise comparison (Fig. 5b), indicating that monomeric Pab1's hydrogen bond network is largely conserved across species, as would be expected for this highly conserved protein.

Generally, various secondary structure types have different tendencies to engage in stable hydrogen bonding, with loops being less hydrogen bonded than helices and sheets which are, by definition, stabilized by amide hydrogen bonding. In the Pab1 structure, the hydrophobic sheets tend to be the most buried away from solvent. Accordingly, we expect that loops will exchange the fastest, followed by helices, and then the stands. This pattern of exchange is seen for the three proteins (under the assumption that the orthologs have the same secondary structure as *S. cerevisiae* Pab1, derived from previous structural work[57]) (Fig. 5c). These results for monomers establish their structural conservation and support our use of the 100 sec as an appropriate time point for making comparison of deuterium uptake levels between the different states.

How similar are the condensate structures across species? To quantify condensate structural similarity across evolutionarily related proteins for the first time (to our knowledge), we calculated the difference in deuterium exchange (Δ%D) for each peptide between the condensed and monomeric proteins (Fig. 5d). Peptides with increased exchange in the condensate have positive Δ%D values, consistent with weaker or a reduced number of hydrogen bonds or increased exposure to solvent in the condensed structure for residues associated with that peptide, and vice versa for peptides with decreased exchange. We assumed unimodal fitting of likely bimodal data to calculate Δ%D which will affect the reported value[56].

We find that the patterns of Δ%D due to condensation are conserved across the Pab1 orthologs (r > 0.72, Fig. 5e). Overall, the four RRMs undergo increased exchange for each ortholog upon condensation, consistent with local unfolding in the condensed structure, as reported for *S. cerevisiae* Pab1[56]. The disordered P domain, in contrast, becomes more protected in each ortholog after condensation (Fig. 5f), consistent with its reported role in *S. cerevisiae*, in providing additional interactions in the condensate context and with the importance of its composition in regulating the temperature of condensation.

The change in the deuterium exchange pattern (Δ%D) at 100 sec upon condensation is similar but not identical for the three orthologs. On average, the change in RRM3 domains in *S. kudriavzevii* is larger than for the other two species, whereas all four RRMs in *K. marxianus* show the smallest difference in exchange between condensate and monomer (Fig. 5f). Taken together, these two observations are consistent with a proposed model for Pab1 condensation where thermally tuned RRM local unfolding is the key event in condensation[56]. It is important to note that the condensates for *S. cerevisiae* and *S. kudriavzevii* were formed at lower temperatures and for different amounts of time (46 °C for 20 min) than *K. marxianus* (46 °C for 20 min, then 50 °C for 10 min, then 55 °C for 10 min). As such, further analysis of the dynamics of these changes is required to fully understand the mechanism of sequence-encoded Pab1 condensation differences among species.

Overall, we find that both Pab1 structure, and structural features of condensates, are largely conserved across three orthologs despite their different condensation onset temperatures. However, notable differences between species among folded domains may explain how

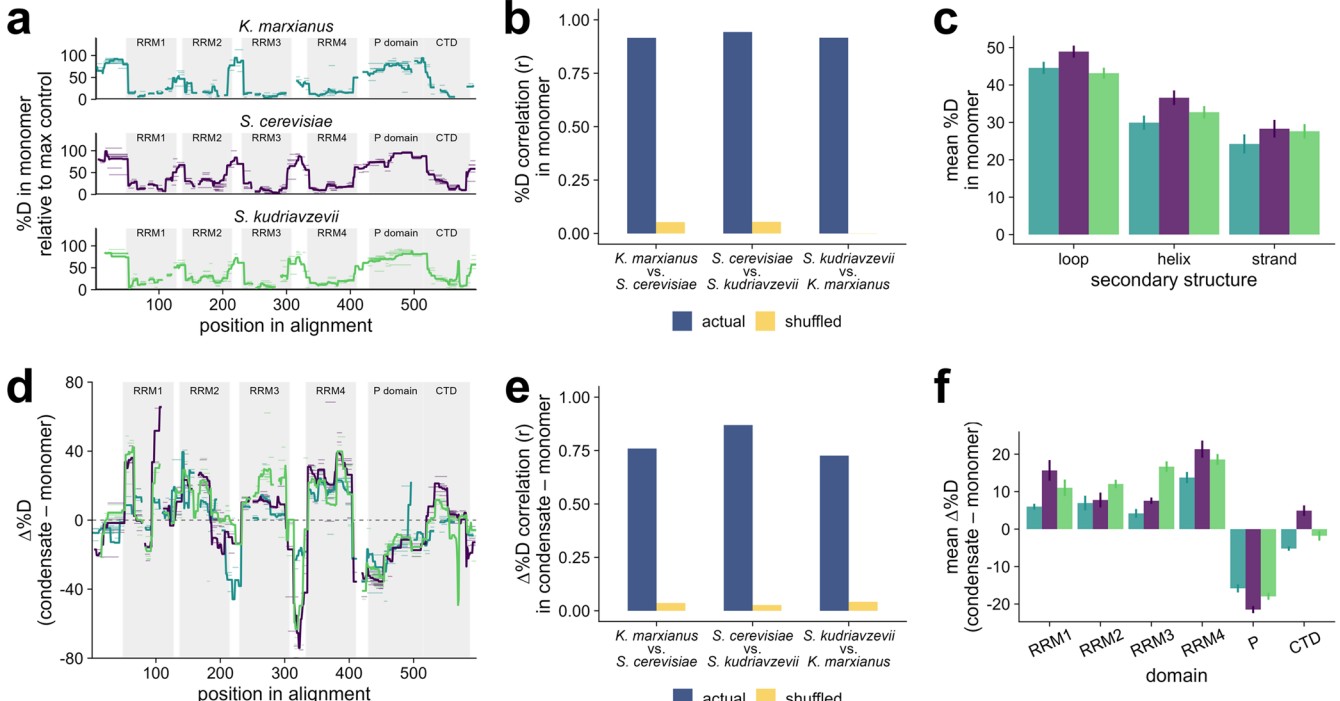

**Fig. 5 | HDX-MS investigation of poly(A)-binding protein (Pab1) ortholog monomers and condensates reveals conservation of condensate structure.**
**a** Pab1 ortholog deuterium (%D) exchange mapped onto aligned primary sequence (all data after 100 seconds of exchange). Domain boundaries are annotated in gray, peptides are light horizontal bars, and solid lines show peptide means at each site.
**b** Correlation (Pearson R) of site-by-site %D exchange for Pab1 monomers between orthologs, with shuffled sites as a control. **c** Relative exchange (%D) in monomers after 100 seconds for secondary structure elements in *S. cerevisiae* Pab1 (PDB: 6R5K). Box and whisker plot is standard: box shows 25th and 75th percentile with median crossbar, whiskers show maxima/minima up to 1.5x the interquartile range, with outliers plotted as separate points. Residue counts for each element type are loop $n = 355$ (368 for *K. marxianus* only), helix $n = 150$, and strand $n = 73$ for all

species. **d** Difference in %D exchange between condensate and monomer for the three Pab1 orthologs, annotated as in **a**. Condensates for *S. cerevisiae* and *S. kudriavzevii* were made by incubating protein at 46 °C for 20 min, *K. marxianus* condensates were made using three incubations: 46 °C for 20 min, then 50 °C for 10 min, then 55 °C for 10 min. **e** Correlation (Pearson R) of differences in %D uptake between condensate and monomer between pairwise species comparisons, with shuffled sites as a control. **f** Change in %D uptake after 100 seconds between condensate and monomer for Pab1 domains. Box and whisker representation is as in **c**. Embedded error bars show mean +/− standard error. For each domain, residue counts for the three species are: RRM1 $n = 68, 75, 69$; RRM2 $n = 78, 71, 65$; RRM3 $n = 69, 75, 55$; RRM4 $n = 75, 78, 78$; CTD $n = 73, 73, 49$. Source data are provided as a Source Data file.

Pab1 condensation can be tuned to an organism's temperature niche. By tuning the stability—and thus activation threshold—of orthologous Pab1 RRMs, nature could enable Pab1 condensation to be triggered in response to each organism's relative stress temperature.

## Discussion

We have measured multiple cellular and molecular responses to temperature in three budding yeast species, diverged by nearly 100 million years, that have adapted to distinct environmental and thermal conditions—the cryophile *S. kudriavzevii*, mesophile *S. cerevisiae*, and thermophile *K. marxianus*. Confirming these designations, cellular growth rate reaches its optimum at different temperatures, and each species' growth ceases at a threshold temperature correlated with this optimum. Upon rapid transfer from optimal growth temperature to near the threshold—heat shock—we observe the classic eukaryotic Hsf1-mediated heat-shock response and sharp downregulation of translation and ribosome biogenesis genes involved in rapid growth in all species. Remarkably, the Msn2/4 regulon, which has long been known to regulate a large fraction of the overall transcriptional response to heat shock and other stresses in *S. cerevisiae*, shows net-zero activation during heat shock in *K. marxianus*.

We carried out what to our knowledge is the first proteome-scale assessment of the conservation of biomolecular condensation across species, examining the condensation response to heat shock which has been previously studied in *S. cerevisiae*[7,11,12,58]. We find that the

biomolecular condensation response is also conserved down to individual protein species, yet tuned to each species' thermal niche.

In principle, this tuned, conserved condensation response could reflect one or more cellular-scale regulatory changes that modulate the behavior of many condensing proteins, such as differences in trehalose production or intracellular pH. Alternatively, individual protein sequences might encode thermal tuning, as has been shown for Ded1[12]. We show that poly(A)-binding protein (Pab1) isolated from each species condenses at the relevant heat shock temperature, indicating that the tuning of these proteins is encoded in the amino acid sequence itself. The adaptive nature of Pab1 condensation during heat stress, which we discovered in *S. cerevisiae*[11], is conserved in *S. kudriavzevii* at the corresponding temperatures for the cryophile. Finally, by comparing structural differences in condensates relative to monomers across species, we are able to propose that temperature-tuned behavior results from changes in RRM stability due to sequence differences in the orthologs.

### Conservation and divergence in the transcriptional response to heat shock

*S. cerevisiae* and *S. kudriavzevii* show similar transcriptome-level responses as expected from their close evolutionary relationship, with divergence 10 million years ago, long after the whole genome duplication in this clade (Fig. 1)[48,59]. The two genomes are so similar that interspecies hybrids can be constructed which display intermediate growth and heat-induced foci phenotypes[60]. *S. cerevisiae* and

*K. marxianus*, which diverged ~100 million years ago and before the whole-genome duplication, show more significant divergence in their transcriptional responses[59]. All three species induce the Hsf1-regulated transcriptional response, as well as downregulate genes involved in protein synthesis and ribosome biogenesis, including transcripts encoding ribosomal protein subunits and those encoding auxiliary biogenesis factors. Responses diverge, however, most notably in the regulation of genes controlled by the transcription factor pair Msn2/4.

The conserved transcriptional responses reflect well-known functional responses to temperature; first, the slowing of growth by suppressing ribosome production and translation factors, and second, massive induction of molecular chaperones in the Hsf1 regulon which respond to temperature-induced protein misfolding and regulate biomolecular condensate formation (Fig. 2). Despite its name, heat shock factor 1 is activated by a wide range of environmental stresses[22,53] during which biomolecular condensation is also observed[4,7,61–66].

Hsf1 is presumably conserved across organisms and across stresses due to the conserved functions of the proteins whose expression it controls, such as molecular chaperones that regulate protein folding and condensation. For example, Hsp70 protein Ssa4, whose transcript is induced by Hsf1 no less than 100-fold at peak across species in our data (and up to 1,000-fold), has been shown to re-localize to condensates in *S. cerevisiae* during heat shock[4]. Hsp70 also plays an essential role in dispersing heat-triggered *S. cerevisiae* Pab1 condensates[15]. Our results reveal the expected—but not previously reported—proteome- and transcriptome-scale correlation between Hsf1 regulon induction and condensation.

Other regulons across the species, however, do not seem to be completely conserved in their heat regulation (Fig. 2). One surprisingly divergent regulon is that of Msn2/4, which has long been considered a master regulator of the environmental stress response[50–53]. Msn2/4-controlled genes are induced in many stresses, and help regulate metabolism and glycolysis[67]. We confirm this standard role in *S. kudriavzevii*, but observe that a substantial fraction of the regulon is downregulated in *K. marxianus* despite the robust Hsf1 response, translation repression, and growth-rate reduction that unambiguously mark the onset of the environmental stress response. We provide further evidence that organisms with varying life histories utilize distinct transcriptional programs to respond to stress[38,47,68], and the *S. cerevisiae* Msn2/4 regulon is incompletely conserved in *K. marxianus*. We speculate this is due to the *Saccharomyces* species' key metabolic difference from the *Kluyveromyces* species, conversion of glucose to ethanol rather than biomass, the so-called Crabtree effect[69].

## Conservation of the biomolecular condensation response to heat shock

The degree to which condensation is conserved across organisms has not been systematically characterized and is a grand challenge in the study of stress granules[70]. Here, we have quantified the conservation of condensation using three closely related species cultured under identical conditions and exposed to stresses differing only by a temperature shift.

Our results provide further evidence that not only is condensation conserved across different niches and over vast timescales, but it is also substantially conserved down to the particular molecular species observed in heat-shock-induced condensates (Fig. 3) and their individual behaviors. For example, poly(A)-binding protein Pab1 drastically decreases its solubility in each organism upon species-specific heat stress, such that a temperature which causes strong condensation of Pab1 from a cryophilic species, 37 °C, is the same temperature which leaves Pab1 from the thermotolerant species fully uncondensed. Previous work has shown this for the RNA helicase Ded1[12]. Here we show that this phenomenon holds proteome-wide: the vast majority of proteins identified as condensing in *S. cerevisiae*[7] have orthologs in *K. marxianus* which also condense, albeit at an elevated temperature. We

also observe that large classes of proteins, such as glycolytic enzymes and ribosomal proteins, are consistently soluble in each species at each species' heat-shock temperature. The broad conservation of condensation behavior provides additional evidence that stress-induced condensation is functionally important, perhaps even serving as a signaling mechanism (as described below) under stressful conditions.

## Conservation and molecular basis of stress-triggered condensation

Many studies have shown that purified components of in vivo condensates will condense autonomously when exposed to stress-related conditions in vitro (reviewed in[71]). For example, translation factor Sup35 forms condensates in yeast cells during glucose starvation, when the intracellular pH drops; in purified form, Sup35 forms gel-like condensates as pH decreases in vitro[72]. Likewise, purified Ded1 from yeast forms heat-induced condensates, and this behavior is conserved and tuned in purified protein from *S. kudriavzevii* as well as the thermophile *T. terrestris*[12]. Purified Pab1's condensation behavior has been shown to be both pH and temperature dependent[11]. Taken together, these results indicate that stress-triggered protein condensation is often encoded directly in the amino acid sequence. Our results support this model, where Pab1 condensation in vitro reflects the organism's thermal niche. We show that conservation goes even further, such that mutations previously shown to shift the condensation temperature of *S. cerevisiae* Pab1 up and down have the same effect in the other two orthologs (Fig. 4).

Using HDX-MS, we show that Pab1 condensates in each species involve similar local conformational changes and new intermolecular contacts at the molecular level (Fig. 5). While we cannot yet provide a detailed mechanistic understanding of how temperature tuning occurs, these protein species and results provide useful tools and strong guidance for such studies. Because Pab1 has been found to condense via a sequential activation and local unfolding mechanism[56], we hypothesize that tuning of the temperature-triggered local unfolding of specific RNA recognition motifs will explain much of the species-specific differences in condensation temperature. Whatever the result, the consistency of local conformational changes across species, but at different thermal baselines, underscores the need for evolutionary explanations rather than purely biophysical constraints.

## Adaptation and evolutionary variation in condensation

Stress-triggered biomolecular condensation in *S. cerevisiae* has been shown to be adaptive, or to have positive regulatory consequences, for specific proteins[11,12,72]. Our results demonstrate that hydrophobicity mutants of Pab1 in *S. kudriavzevii* mirror what has been observed in *S. cerevisiae*: suppressing temperature-induced condensation of Pab1 by increasing its condensation temperature reduces the ability of the organism to grow at high temperature (Fig. 4). This results from a reversible growth arrest, rather than decreased survival, at high temperatures (Figure S1). What is remarkable about this result is that "high temperature" for *S. kudriavzevii* is very nearly the optimal growth temperature for *S. cerevisiae*—the same adaptive phenotypic consequences appear in the cryophile, but at a temperature eight degrees lower. These evolved adaptive behaviors further support rethinking the response to heat shock, expanding from the longstanding focus on deleterious protein misfolding and proteotoxic aggregation to encompass large-scale adaptive reorganization by biomolecular condensation. Specifically, we propose that condensation acts as a signal transduction mechanism, converting temperature information into system-scale changes in molecular structure and organization with exquisite sensitivity[73]. Such condensation-mediated signal transduction integrates cleanly into the established model for transcriptional activation of Hsf1 by titration of chaperones—now well-established[13,14]—to condensates, as proposed[11] and demonstrated[15]. Given the massive number of condensing protein species, a wide range

of other potential functions are possible—and, we argue, likely given the scale of conservation we observe.

From ecological niche, to patterns of growth and survival, to gene expression and proteome-scale biomolecular condensation, to the condensation behavior of purified molecular species, to molecular changes, we show that cellular responses to stress are exquisitely preserved and precisely tuned over vast evolutionary timescales so that organisms can flourish in distinct thermal niches. The systematic characterization of these three thermally adapted fungi reported here not only demonstrates this tuning, but also provides a wealth of large-scale information essential for moving beyond anecdotal studies to a deeper understanding of the principles underlying variation in condensation across the tree of life.

## Methods

### Identification of orthologs and divergence times

Orthologs between *S. cerevisiae* and *S. kudriavzevii* were found using the Yeast Gene Order Browser v8 (http://ygob.ucd.ie/) using Pillars.tab file to confidently assign homology[74]. *K. marxianus* orthologs to *S. cerevisiae* were obtained using both the KEGG BRITE Orthology database[75] and the eggNOG v5.0 database[76]. Ortholog assignments used in this study can be found in Supplementary File 1. Divergence times were estimated using timetree.org[59].

### Maximum specific growth assays

Three wild-type strains were used to measure growth: *S. cerevisiae* BY4742, *S. kudriavzevii* FM1389, and *K. marxianus* DMKU3-1042. A dense overnight culture was diluted into YPD and allowed to grow to $OD_{600}$ ~ 0.1. $OD_{600}$ measurements were taken periodically during log phase growth. At least two biological replicates were performed for each species and time point. Maximum specific growth rates were obtained by estimating the slope of the linear range of growth for each species. Resulting growth curves were fit using the cardinal temperature model with inflection[77].

### Strain construction

*Saccharomyces kudriavzevii* mutants were obtained using CRISPR-Cas9 genome editing. Briefly, competent FM1389 were prepared (Frozen-EZ Yeast Transformation II Kit™, Zymo Research). Competent cells were transformed according to the protocol except with a 90 min incubation at 23 °C. DNA was used at concentrations of at least 200 μg of Cas9/guide RNA-containing plasmid with a *URA3* selectable marker along with >500 μg of linear repair template as previously described[78,79]. Transformation reactions were spread on -Ura plates. Transformants were cured of the Cas9/guide RNA plasmid and sequence confirmed. Multiple transformants obtained using distinct guide RNAs were screened to check for phenotypic homogeneity.

A plasmid encoding SSA3p-eGFP (pSK190) was transformed into *K. marxianus* RAK3877 (DMKU3-1042 *ura3-1 his5-1*) according to previously published protocol[80].

### Spot assay

For each strain, a dense overnight culture was diluted into fresh YPD and allowed to grow for at least 6 hours until cells reached $OD_{600}$ of at least 0.1. Each culture was diluted to a matching $OD_{600}$. Cultures were then serially diluted 10-fold into $dH_2O$, and 7 μL of each dilution was spotted onto plates. Plates were incubated at the specified temperature for 4 days and then imaged. Plates were then moved to room temperature and incubated for 2 days and imaged again.

### Flow cytometry

Cells were grown overnight to $OD_{600}$ ~ 0.05 in SC-complete medium with 2% dextrose, temperature treated for 20 minutes, and recovered for 3 hours with shaking at 23 °C. Data were collected on the AttuneNxT (Thermo Fisher) at 100 uL/min. At least 20,000 events were recorded per 100 uL of cells. Voltage was set as follows: Forward scatter: 1; Side scatter: 200; YL2 (mCherry): 540; VL1 (autofluorescence): 400.

All experiments were performed with the same voltage set, and the fluorescence values reported reflect forward scatter area and autofluorescence-normalized values. Subpopulations of cells that exhibited high autofluorescence (abnormally high BL2 signal) were removed from the analysis; at least 5,000 cells per sample remained. Fold-change values were calculated from mock-treated cells grown at 23 °C.

Fitting of skew-normal distributions for heat shock response data (Fig. 1c) were carried out using R[81] and the 'sn' package[82].

### RNA sequencing

**Sample preparation**. Each species (strains BY4742, FM1389, DMKU3-1042) was grown at its optimum growth temperature until $OD_{600}$ ~ 0.2-0.4. 1.2 mL of cells were transferred to a 1.5 mL tube and treated at the specified temperature for 8 minutes. Cells were centrifuged at 3,000 g for 1 minute at room temperature, and the supernatant was removed. Cells were flash frozen and stored at −80 °C.

**Library generation**. Total RNA was extracted using Direct-zol RNA Miniprep kit (Zymo cat. no R2052) with Ambion TRI-reagent (Fisher cat. no. AM9738) and Zirconia/Silica 0.5 mm homogenization beads (Biospec cat. no. 11079105Z). Samples were treated with Dnase I (NEB M0303S). Two biological replicates were generated for each organism and condition. Sequencing libraries were prepared by Novogene (https://www.novogene.com/us-en/). Messenger RNA was purified from total RNA using poly-T oligo-attached magnetic beads. After fragmentation, the first strand cDNA was synthesized using random hexamer primers followed by the second strand cDNA synthesis, end repair, A-tailing, adapter ligation, size selection, amplification, and purification. Paired-end sequencing of the library was performed on an Illumina instrument.

**Data processing**. RNA sequencing reads were processed using a custom Snakemake pipeline available here: https://github.com/drummondlab/conservation-of-condensation-2024/[83]. Raw reads were first processed using TrimGalore v0.6.10 to remove Illumina sequencing adapters using default settings (doi: 10.5281/zenodo.7598955)[84]. All genomic DNA sequences and GTF files were downloaded from NCBI using the following versions (*Saccharomyces cerevisiae*: GCF_000146045.2_R64, *Saccharomyces kudriavzevii*: GCA_947243785.1_Skud-ZP591, *Kluyveromyces marxianus*: GCA_001417885.1_Kmar_1.0). These files were then used by STAR v2.7.10b to generate indices (--sjdbOverhang 99 --sjdbGTFtagExonParentTranscript transcript_id --sjdbGTFfeatureExon CDS --sjdbGTFtagExonParentGene gene_id --genomeSAindexNbases 10)[85]. Mapping of the filtered and trimmed fastqs was also done with STAR v2.7.10b (--alignMatesGapMax 20000). The reads mapping to each gene were quantified using HTSeq v2.0.2 (--stranded=no --type=CDS --idattr=gene_id)[86].

**Data analysis**. Gene lengths were extracted for each gene by first adding exon annotations to the GTF files using a custom script based on gffutils v0.11.1 (https://github.com/daler/gffutils). Gene lengths were then calculated using the GenomicFeatures package in R[87]. These lengths were then used to calculate transcript per million values (TPMs). In addition, fold changes in transcript abundance were calculated using DESeq2 v3.16[88].

**Gene annotation**. Protein annotations listed in Supplementary File 2 were derived from the following sources. The targets of HSF1 and Msn2/4 were curated from[3] and[2]. The genes for core ribosomal proteins, ribosome biogenesis factors, and glycolytic enzymes

(superpathway of glucose fermentation) as well as transcription factor regulation assignments were derived from the Saccharomyces Genome Database[89,90]. Genes for translation factors were derived from the KEGG BRITE database[91]. Transcription factor regulators in Supplementary File 3 were assigned according to[16].

## In vivo biochemical fractionation and mass spectrometry sample preparation

Cells (BY4743, FM1171, or DMKU3-1042) were grown in SC-complete medium with 2% dextrose (50 mL of medium per treatment) at either 23 °C or 30 °C with 250 rpm shaking until $OD_{600}$ was between 0.4-0.6. Cells were transferred to a 50 mL conical tube and harvested via centrifugation at 2500 g for 5 min in a swinging bucket rotor at room temperature. The supernatant was decanted and the conical tubes containing cells were immediately placed in water baths set to the treatment temperature. After 8 minutes, cells were resuspended in 1 mL ice-cold soluble protein buffer (SPB from[7]), transferred to a pre-chilled 1.5 mL tube, and spun for 30 s at 5000 g at 4 °C. The supernatant was discarded, and the cell pellet was resuspended in 200 µL of SPB. Two 100 µL aliquots from the resuspended sample were snap-frozen in a safe-lock tube.

Cells were lysed using cryomilling and fractionated with ultracentrifugation as described in[7], with minor modifications. Briefly, cells were lysed with 5 × 90 s agitations at 30 Hz. After lysis, cellular material was resuspended in 900 µL of SPB and thawed on ice with occasional vortexing. The lysate was clarified to remove very large aggregates and membrane components with a 3000 g spin for 30 s at 4 °C. Then, 650 µL of clarified lysate was moved to a pre-chilled 1.5 mL tube. For a Total sample, 100 µL was mixed with 300 µL total protein buffer (TPB from[7]) and processed as described in[7]. Of the remaining clarified lysate, 500 µL was transferred to a vacuum safe tube spun at 100,000 g for 20 minutes at 4 °C (fixed-angle TLA-55 rotor in a Beckman Coulter Optimax tabletop ultracentrifuge). The supernatant fraction was decanted (as much as 400 µL) and snap frozen. The pellet fraction was isolated as described in[7].

Protein was extracted from each fractionated sample using a chloroform-methanol method adapted from[92]. To 100 µL of sample, 400 µL of methanol was added and vortexed. Then 100 µL of chloroform was added and vortexed. 300 µL of $dH_2O$ was added and samples were vortexed. Samples were centrifuged for 1 minute at 14,000 g. The top aqueous layer was removed, and 400 µL of methanol was added. Samples were vortexed and then centrifuged for 5 minutes at 20,000 g. The remaining methanol was removed with a pipette, and the sample was dried at 55 °C. The protein flake was submitted for digestion and mass spectrometry at MS Bioworks.

## Mass spectrometry

**Sample preparation.** Protein flake was solubilized in urea lysis buffer (8 M urea, 50 mM Tris.HCl pH8, 150 mM NaCl, 1X Roche cOmplete protease inhibitor) using a QSonica sonic probe with the following settings: Amplitude 50%, Pulse 10 x 1s. 1 on 1 off. The lysate was incubated at room temperature for 1 hr with mixing at 1000 rpm in an Eppendorf ThermoMixer and then centrifuged at 10,000 g for 10 minutes at 25 °C. Protein quantitation was performed using a Qubit protein assay (Invitrogen), protein yields are provided below. 25 µg of each lysate was digested as follows: reduced with 15 mM dithiothreitol at 25 °C for 30 minutes followed by alkylation with 15 mM; iodoacetamide at 25 °C for 45 minutes in the dark; digested with 2.5 µg sequencing grade trypsin (Promega) at 37 °C overnight. The final digest volume was 0.5 mL adjusted with 25 mM ammonium bicarbonate; cooled to 25 °C, acidified with formic acid and desalted using a Waters Oasis HLB solid phase extraction plate; eluted samples were frozen and lyophilized; pooled sample was made by mixing equal amounts of digested material from each sample. This pooled sample was used to generate a gas phase fractionation library.

**DIA chromatogram library generation.** 1 µg of the pooled sample was analyzed by nano LC- MS/MS with a Waters M-class HPLC system interfaced to a ThermoFisher Exploris 480. Peptides were loaded on a trapping column and eluted over a 75µm analytical column at 350nL/min; both columns were packed with XSelect CSH C18 resin (Waters); the trapping column contained a 5µm particle, the analytical column contained a 2.4µm particle. The column was heated to 55 °C using a column heater (Sonation). The sample was analyzed using 6 × 1.5 h gradients. Six gas-phase fractions (GPF) injections were acquired for 6 ranges: 396 to 502, 496 to 602, 596 to 702, 696 to 802, 796 to 902, and 896 to 1002. Sequentially, full scan MS data (60,000 FWHM resolution) was followed by 26 × 4 m/z precursor isolation windows, another full scan and 26 ×4 m/z windows staggered

by 2 m/z; products were acquired at 30,000 FWHM resolution. The automatic gain control (AGC) target was set to 1e6 for both full MS and product ion data. The maximum ion inject time (IIT) was set to 50 ms for full MS and dynamic mode for products with 9 data points required across the peak; the NCE was set to 30.

**Sample analysis.** Samples were randomized for acquisition. 1 µg per sample was analyzed by nano LC/MS with a Waters M-class HPLC system interfaced to a ThermoFisher Exploris 480. Peptides were loaded on a trapping column and eluted over a 75µm analytical column at 350nL/min; both columns were packed with XSelect CSH C18 resin (Waters); the trapping column contained a 5 µm particle, the analytical column contained a 2.4µm particle. The column was heated to 55 °C using a column heater (Sonation). Samples were analyzed using 1.5 h gradients. The mass spectrometer was operated in data-independent mode. Sequentially, full scan MS data (60,000 FWHM resolution) from m/z 385-1015 was followed by 61 × 10 m/z precursor isolation windows, another full scan from m/z 385-1015 was followed by 61 ×10 m/z windows staggered by 5 m/z; products were acquired at 15,000 FWHM resolution. The maximum ion inject time (IIT) was set to 50 ms for full MS and dynamic mode for products with 9 data points required across the peak; the NCE was set to 30.

**Data processing.** Data are available via ProteomeXchange with identifiers PXD044702. DIA data were analyzed using Scaffold DIA 3.3.1 (Proteome Software) which served several functions: 1) Conversion of RAW files to mzML (ProteoWizard) including deconvolution of staggered windows; 2) Conversion to DIA format; 3) Alignment based on retention times; 4) Searching of data using Prosit library (DLIB) and the chromatogram/reference library to create custom ELIB; 5) Filtering of database search results using Percolator at 1% peptide false discovery rate; 6) Calculation of peak areas for detected peptides using Encyclopedia (0.9.6). For each peptide the 5 highest quality fragment ions were selected for quantitation.

Data were searched with the following parameters: Enzyme: Trypsin; Database: UniProt *Kluyveromyces marxianus*, *Saccharomyces kudriavzevii*, or *Saccharomyces cerevisiae*; Fixed modification: Carbamidomethyl (C); Precursor Mass Tolerance: 10ppm; Fragment Mass Tolerance: 10ppm; Library Fragment Tolerance: 10ppm; Peptide FDR: 0.01; Protein FDR: 0.01; Peptide Length: 6-30AA; Max Missed Cleavages: 1; Min. Peptides: 2; Peptide Charge: 2-3.

The samples table was exported from Scaffold DIA and further analyzed in Perseus[93] and R[81].

**Calculation of pSup.** The statistical model used to estimate the proportion in supernatant (pSup) was based on that used in[7]. For each fractionated sample, the relative abundance of proteins within each fraction—total (T), supernatant (S), and pellet (P)—were inferred from mass spectrometric data. While proteins are expected to obey conservation of mass in the original fractionated lysate ($T_i = S_i + P_i$ for protein species i), this assumption does not hold in the ratios of abundances directly inferred from the data. Instead, for a particular

experiment, $T_i = a_S S_i + a_P P_i$ where we refer to the per-experiment constants $a_S$ and $a_P$ as mixing ratios which reflect differential processing and measurement of individual fractions. In order to estimate mixing ratios, and thus recover the original stoichiometry, we assume conservation of mass for each protein in the sample, and then use Markov Chain Monte Carlo sampling to estimate the mixing ratios under this constraint[94]. We also assume negative binomial noise for each measurement. Specifically, we model mRNA abundance as follows:

$$\log(T_i) \sim N(\log(\alpha_S S_i + \alpha_P P_i), \sigma)$$

where

$T_i$ = measured abundance of protein $i$,
$S_i$ = measured abundance in supernatant of protein $i$,
$P_i$ = measured abundance in pellet of protein $i$,
$a_s$ = mixing ratio of supernatant sample,
$a_p$ = mixing ratio of pellet sample
With the following priors:

$$\alpha_S \sim \Gamma(1,1)$$

$$\alpha_P \sim \Gamma(1,1)$$

$$\sigma \sim Cauchy(0,3)$$

We implemented the model above in R using the probabilistic programming language STAN, accessed using the rstan package[81,95] and used all proteins with *intensity* > 1 to estimate mixing ratios for each sample. These mixing ratios were then used to calculate the pSup for protein i: $pSup_i = \frac{\alpha_S S_i}{\alpha_S S_i + \alpha_P P_i}$.

### Pab1 wild-type and mutant protein purification
Pab1 wild-type and mutant proteins were purified according to[15] with one modification. The sizing column used was a Superose 6 Increase 10/300 GL (GE Healthcare).

### Dynamic light scattering of purified protein
DLS measurements were performed in a DynaPro NanoStar Plate Reader (Wyatt). Each time point was the average of five 6-second acquisitions. Measurements were performed at a slow ramp (0.25 °C/min) starting at 25 °C and ending at 55 °C. All experiments were performed with 30 μL of 15 μM purified Pab1 protein, dialyzed overnight in freshly prepared buffer with 20 mM MES pH 6.4, 150 mM KCl, 2.5 mM MgCl$_2$, and 1 mM DTT. Samples were centrifuged for 20 min at 20,000 g at 20 °C before loading in the plate. To prevent evaporation of the sample, 10 μL of mineral oil was layered on top of the sample. $T_{condense}$ is calculated as previously described as $T_{demix}$[11] except that the value of the baseline was calculated as the average R$_h$ values below 35 °C. This change was implemented to account for the fewer acquisitions per sample due to use of the plate reader rather than a single cuvette. Calculated monomer baseline sizes and $T_{condense}$ temperatures can be found in Supplementary Tables 1 and 2, respectively.

### Wide-field microscopy
**Dye labeling of proteins.** Purified Pab1 of each species (*S. cerevisiae, S. kudriavzevii, K. marxianus*) was buffer exchanged (Zeba spin columns, 7 K MWCO) into a pH 8.2 amine-free labeling buffer (phosphate buffered saline solution supplemented with 0.1 M sodium bicarbonate) and reacted with ATTO 647N NHS ester (ATTO-TEC) for 70 minutes at room temperature while shaking at 300 rpm. Following the dye reaction, Pab1 was buffer exchanged to 20 mM MES pH 6.4, 150 mM KCl, 2.5 mM MgCl$_2$, and 1 mM DTT. The degree of labeling for all Pab1 species was 0.3 (dye to protein molar ratio), as measured by

absorbance ratio with correction for dye absorbance at A280 (Nano-Drop One, ThermoFisher).

**Preparation of samples for imaging.** Each labeled protein sample was mixed with unlabeled wild-type protein and buffer (20 mM MES pH 6.4, 150 mM KCl, 2.5 mM MgCl$_2$, and 1 mM DTT) to a final concentration of 10 μM unlabeled Pab1 and 0.2 μM labeled Pab1 (50:1 unlabeled:labeled). The Pab1 solution (50 uL) was transferred to a custom imaging chamber consisting of a plasma-cleaned imaging substrate compatible with on-microscope temperature control (VAHEAT, Interherence) and a 0.9 mm depth perfusion chamber (Grace Bio-Labs).

**Fluorescence imaging with on-microscope temperature control.** Images were collected on a custom wide-field imaging setup illuminated at 637 nm (OBIS LX, Coherent) with an sCMOS detection system (Photometrics Prime 95B). Rapid temperature control of samples was achieved with a VAHEAT temperature control unit (Interherence). Samples were initially brought to 30 °C for four minutes, then incrementally heated to higher temperatures at four-minute intervals (37 °C, 40 °C, 42 °C, 46 °C, 50 °C, 55 °C). Samples stabilized at each new temperature within seconds. Data were acquired in the last two minutes of each temperature treatment (at least two minutes after changing temperature). For each species and temperature condition, multiple image stacks of 100 frames each were taken at 100 ms exposure time and with approximately 1 W/cm$^2$ of continuous 637 nm excitation. For each condition, a representative frame was selected. All frames were cropped in ImageJ to the same area at the same pixel positions. Three sets of colorbar scalings (2500–6000, 2500–18000, 2500–30000, as visually indicated in Figure S4) were required to visualize condensates from different species at different temperatures.

### Hydrogen-deuterium exchange mass spectrometry
**Condensate preparation.** Condensates were largely prepared as in[56]. A 10 μM Pab1 stock was exposed to the condensing condition at pH 6.5 (*S. cerevisiae* or *S. kudriavzevii* Pab1 at 46 °C for 20 min, and *K. marxianus* Pab1 at 46 °C for 20 min, followed by 50 °C for 10 min, followed by 55 °C for 10 min). After temperature exposure, condensates were collected via a 16,000 g spin for 10 minutes. The supernatant was discarded, and the pellets were washed twice using the same centrifugation procedure.

**HDX labeling.** HDX labeling was completed as in[56] with the following exceptions: first, for both monomer and condensed states for the 3 orthologs, samples at HDX timepoints of 100, 400, 800, 1500, 3000, and 12900 seconds were collected with pD$_{corr}$ 6; second, "saturated" control samples were labeled with pD$_{corr}$ 6 overnight.

**LC-MS/MS and HDX-MS data analysis.** LC-MS/MS was completed as previously described[96]. HDX-MS data analysis was completed as in[56], with the exception that peptides were filtered by 'Medium' or 'High' confidence fitting from HDExaminer and downstream analysis was completed using R. Data are available via ProteomeXchange with identifiers PXD044970.

**Peptide maps.** The peptide maps were largely generated as previously described[56]. Briefly, the assignment was completed by searching the MS/MS data of each homolog against a database containing the sequences of each Pab1, the proteases, and all other proteins running on the LC-MS system. This was done using SearchGUI software[97] with the following search settings: unspecific cleavage, precursor charge 1-4, isotopes 0-1, precursor m/z tolerance 10.0 ppm, fragment m/z tolerance 10.0ppm, no post-translational modifications, peptide length 8-30. These search results were imported into PeptideShaker[98] before undergoing further processing using EXMS2 software[99] to generate peptide lists and visualize peptide maps.

**Table 1 | Strains and plasmids**

| Organism/strain | Genotype | Source |
|---|---|---|
| S. cerevisiae BY4742 | MATα ura3Δ0 leu2Δ0 his3Δ1 lys2Δ0 | 103 |
| S. kudriavzevii FM1389 (original ZP591) | MATα hoΔ::kanMX ura3Δ0 his3Δ0 | 104 |
| K. marxianus DMKU3-1042 | MATα | NBRC 104275 (Japan) |
| K. marxianus RAK3877 | MATα ura3-1 his5-1 | 105 |
| S. cerevisiae yCDK061 | MATα ura3Δ0 leu2Δ0 his3Δ1 lys2Δ0 PAB1-1255-1506::Pab1*1255-1506, MV → A | 11 |
| S. cerevisiae yCDK066 | MATα ura3Δ0 leu2Δ0 his3Δ1 lys2Δ0 PAB1-1255-1506::Pab1*1255-1506, MV → I | 11 |
| S. kudriavzevii ySMK103 | MATα hoΔ::kanMX ura3Δ0 his3Δ0 SKDZ_05G2450(PAB1)–1255-1506::SKDZ_05G2450(Pab1)*1255-1506, MV → A | This study |
| S. kudriavzevii ySMK105 | MATα hoΔ::kanMX ura3Δ0 his3Δ0 SKDZ_05G2450(PAB1)–1255-1506::SKDZ_05G2450(Pab1)*1255-1506, MV → I | This study |
| S. kudriavzevii FM1171 | MATa/MATα | 106 |
| S. cerevisiae BY4743 | MATa/α ura3Δ0/ura3Δ0 leu2Δ0/leu2Δ0 his3Δ0/his3Δ0 MET15/met15Δ0 LYS2/lys2Δ0 | 103 |
| S. cerevisiae yCGT027 | MATα ura3Δ0 leu2Δ0 his3Δ1 lys2Δ0 SSA4::SSA4-mCherry | 16 |
| S. kudriavzevii ySMK009 | MATα hoΔ::kanMX ura3Δ0 his3Δ0 SKDZ_05G1830(SSA4)::SKDZ_05G1830(SSA4)-mCherry | This study |
| **Plasmid** | **Details** | **Source** |
| pSK166 | 8xHis-(TevC)-SKDZ_05G2450(PAB1) in pET28a backbone | This paper |
| pSK168 | 8xHis-(TevC)-SKDZ_05G2450(PAB1) MV → A in pET28a background | This paper |
| pSK170 | 8xHis-(TevC)-SKDZ_05G2450(PAB1) MV → I in pET28a background | This paper |
| pSK172 | pV1382 + SKDZ_05G2450(PAB1) P-domain guide CGGCCGCAGCGGTAGCTTGT | 107 this paper |
| pSK173 | pV1382 +SKDZ_05G2450(PAB1) P-domain guide CAACAAATGAACCCAATGGG | 107 this paper |
| pSK174 | pV1382 + SKDZ_05G2450(PAB1) P-domain guide TACGGTGTCCCTCCACAAGG | 107 this paper |
| pSK180 | 8xHis-(TevC)-KLMA_30299(PAB1) wild type in pET28a background | This paper |
| pSK181 | 8xHis-(TevC)-KLMA_30299(PAB1) MV → A in pET28a background | This paper |
| pSK182 | 8xHis-(TevC)-KLMA_30299(PAB1) MV → I in pET28a background | This paper |
| pSK022 | pCGT07 + SKDZ_05G1830(SSA4)- mCherry repair template carrier | 16 this study |
| pSK025 | Cen6-URA + Cas9-PGK1 + SKDZ_05G1830(SSA4) guide CGGAACCGGACCTGCTCCAG | 78 this paper |
| pSK026 | Cen6-URA + Cas9-PGK1 + SKDZ_05G1830(SSA4) guide GGTGCTGGAGCAGGCCCAAG | 78 this paper |
| pJAR006 | 8xHis-(TevC)-Pab1 in pET28a backbone | 11 |
| pJAR033 | 8xHis-(TevC)-Pab1 MV → A in pET28a background | 11 |
| pJAR035 | 8xHis-(TevC)-Pab1 MV → I in pET28a background | 11 |
| pSK190 | pSSA3-eGFP-CYC1t with Kmarx CEN/ARS, HIS5 | 80 this paper |

### Code and data analysis

All data analysis was performed with R[81] using packages from the tidyverse[100] and others as indicated in the methods section. Plots were made with ggplot2[101] or UpSetR[102]. Custom R packages can be found on GitHub (https://github.com/ctriandafillou/flownalysis; https://github.com/ctriandafillou/cat.extras). RNA sequencing reads were processed using a custom Snakemake pipeline available here: https://github.com/drummondlab/conservation-of-condensation-2024/[83]. Raw data and scripts that produce plots that appear in this work are available on Dryad (https://doi.org/10.5061/dryad.w3r2280w6).

### Strains and plasmids

Strains and plasmids used in this study are listed in main text Table 1.

### Statistical tests

All correlations were calculated in R using Spearman's or Pearson's correlation coefficient, as indicated.

### Reporting summary

Further information on research design is available in the Nature Portfolio Reporting Summary linked to this article.

### Data availability

The yeast proteome is available from Saccharomyces Genome Database http://sgd-archive.yeastgenome.org/sequence/S288C_reference/orf_protein/. Sequencing data have been deposited in GEO under accession code GSE234499. The mass spectrometry proteomics data have been deposited to the ProteomeXchange Consortium via the PRIDE[108] partner repository with the dataset identifiers PXD044702 and PXD044970. Raw and processed flow cytometry data, HDX-MS data, growth data, TSP-LC-MS/MS data, imaging data, and DLS data to reproduce all figures have been deposited to Dryad (https://doi.org/10.5061/dryad.w3r2280w6) or GitHub (https://github.com/drummondlab/conservation-of-condensation-2024). Source data are provided with this paper.

### Code availability

Code to reproduce all figures has been deposited on GitHub (https://github.com/drummondlab/conservation-of-condensation-2024).

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

## Acknowledgements

We thank Kevin Byrne, L.G. Macías, D. Peris, A.A.R. Martos, C. Toft, C.T. Hittinger, E. Barrio, and K.H. Wolfe for graciously sharing *S. kudriavzevii* annotations; Ian Wheeldon and Sangcheon Lee from the University of California-Riverside for sharing knowledge and plasmids for *K. marxianus*; David Pincus and Asif Ali for helpful comments and collaborations with previous iterations of this project; and Edward Wallace and Xuejia Ke for assistance with modeling and coding. We thank Dr. Hisashi Hoshida from Yamaguchi University and the Research Center for Thermotolerant Microbial Resources at Yamaguchi University (YU-RCTMR) for sharing auxotrophic *K. marxianus* strains. We acknowledge valuable assistance from the Biophysics Facility and the Sequencing Core Facility at the University of Chicago. D.A.D. acknowledges support from NIH (award GM144278 and GM127406). T.R.S. acknowledges support from NIH (award R01 GM055694, R35 GM14833). S.K.K. acknowledges support from NIH (award T32 GM007197-43 and F31 ES032337-01). H.G. acknowledges support from NIH (award F30 ES032665). K.M.L. acknowledges support from NIH (awards T32 GM007281-45 and F30 ES035279-01), a Frank Family Fellowship, and the Dr. Kenneth S. Polonsky Fellowship. A.H.S. acknowledges support from NSF (award OMA-2121044) and the Neubauer Family Foundation. J.A.M.B. acknowledges support from the Helen Hay Whitney Foundation. The content is solely the responsibility of the authors and does not necessarily represent the official views of the NIH.

## Author contributions

Conceptualization, D.A.D.; methodology S.K.K., D.C., C.W.H., H.G., J.A.M.B., M.F., T.R.S., and D.A.D.; investigation, S.K.K., D.C., C.W.H, H.G., M.F., and K.M.L.; formal analysis, S.K.K., D.C., C.W.H., H.G., J.A.M.B., K.M.L., A.H.S., and D.A.D.; visualization, S.K.K. and D.A.D.; writing - original draft, S.K.K. and H.G.; writing - review & editing, S.K.K., D.C., C.W.H., H.G., J.A.M.B., M.F., T.R.S., and D.A.D.; funding acquisition, S.K.K., T.R.S., and D.A.D.; resources, T.R.S. and D.A.D.; supervision, T.R.S. and D.A.D.

## Competing interests

The authors declare no competing interests.
