## [Peer Review File · Nature Communications]

An adaptive biomolecular condensation response is conserved across environmentally divergent speciesREVIEWER COMMENTS

Reviewer #1 (Remarks to the Author):

In the manuscript “An adaptive biomolecular condensation response is conserved across environmentally divergent species,” Kik et al investigate the evolutionary conservation of the stress-triggered condensation proteome. They focus on comparing responses in related *Saccharomyces* species chosen to represent different thermal niches and also genome duplication status. They compare the temperature-dependent changes in the transcriptomes and condensed-phase proteomes to assess global changes in cellular composition in response to heat shock. They first establish that the three species show heat shock responses at different temperatures that in each case are few degrees higher than the temperature at which they grow maximally. They observe that Hsp70 response and changes in Hsf1 regulated genes show similar response patterns albeit at the corresponding different heat shock temperatures for the three species. While the pre- whole genome duplication species *K. marxianus* deviates from the transcript abundance trends for Msn2/4 target orthologs upon heat shock seen in the two other species, the authors attribute this difference to its respiratory lifestyle. Except for these targets, targets of other transcription factors appear to show an evolutionarily-conserved transcriptional response.

The authors then use LC-MS/MS of fractionated lysates upon heat shock to evaluate changes in solubility of proteins. They report that many orthologous proteins known to show heat-induced condensation, in particular those that have been described as super-aggregators lose solubility upon heat shock and do so at the species-specific heat shock temperatures. In contrast glycolytic proteins remain soluble suggesting this is not a general denaturation and aggregation phenomenon. Next the authors compare the characteristics of Pab1 condensation across the test species building off the models of Pab1 function the group has described in *S. cerevisiae*. They examine the impact of hydrophobic residues on the temperature sensitivity of Pab1 by making mutating them to alter their hydrophobicity. They report that all changes that decrease hydrophobicity similarly decrease the temperature threshold for condensation. They see the opposite trend with increased hydrophobicity. Reduced hydrophobicity also impacted fitness under heat shock, suggesting that condensation of Pab1 promotes heat shock tolerance in *S. cerevisiae* and *S. kudriavzevii*. Finally they investigate the structural basis of Pab1 condensation using deuterium exchange MS to examine the structural dynamics of different regions of Pab1.

The authors conclude that stress-triggered condensation is an evolutionary conserved adaptive sensing program upstream of transcriptional response.

The work is thorough and presented clearly. Although it doesn't add substantially to a mechanistic understanding of the stress-response, it shows evolutionary conservation of the mechanism. While I wouldn't require this in a revision, I am curious about the recovery from stress and if the reverse steps are comparable conserved. Are the super aggregators reversible upon return to permissive temperature?

It would be interesting to perform the proteomics on the soluble and insoluble fractions following a recovery. In general, this is a well done, well written set of experiments that is an important example of how evolutionarily divergent organisms can be a useful lens to understand the process of stress adaptation and biomolecular condensation. This approach is especially important as the field grapples with decoding the molecular grammar of IDRs.

Reviewer #2 (Remarks to the Author):

In this manuscript, Samantha Keyport Kik and colleagues used three evolutionarily diverged budding yeast species adapted to different temperatures (cryophilic, mesophilic, and thermotolerant) to understand whether protein condensation may act as a sensor of and coordinate the stress responses at the transcriptional and translational levels. The manuscript is well-thought, well-organized and addresses an important question.

There are two aspects that may deserve more detailed investigation and discussion.

First, using purified Pab1 from the three different species, which show 69-98% sequence identity, the authors could show that temperature sensitivity and condensation at a specific temperature is encoded in the primary protein sequence, similar to what already published for Ded1. The authors identify a group of proteins, called “superaggregators”, that condense and lose solubility upon heat shock. Do superaggregators include proteins that play a primary role in the process of condensation and proteins that act as clients and are recruited inside forming condensates (proteins that co-condense)? Or do they exclude proteins known to co-condense (without triggering by themselves the formation of condensates)? It would be informative, based on the existing literature, to stress the difference between these two types of proteins (condensing and co-condensing), in order to verify whether the optimal temperature required for condensation does depend on the exact sequence also for the co-condensing proteins or only for the condensing ones.

Second, by comparing structural differences in condensates relative to monomers using HDX-MS, temperature-dependent condensation and growth rate the authors conclude that condensation *in vivo* is required for an organism to grow at temperatures above that of optimal growth. Under certain experimental conditions, growth was not resumed during the stress recovery phase, likely due to death (Figure 4). Does death correlate with the formation of irreversible condensates? And does cell growth recovery occur when condensates dissolve during the stress recovery phase? It will be important to address this point, given the accumulating evidence showing that cells ability to respond and adapt to acute environmental stress, including heat shock and acidosis, depends on the formation of reversible

amyloidogenic condensates (that are dispersed with the assistance of the protein quality control system) (PMID: 27720612; PMID: 34704593; PMID: 31271238).

Other comments:

Figure 4: Dynamic light scattering (DLS) and the increases in Rh have been used here to identify the onset of condensation, as previously established by the authors (Riback et al., Cell 2017). The authors should consider adding representative images showing the Pab1 condensates (WT and mutants) at the different temperatures and in the different species.

Minor comments

"Nevertheless, the HSR can be robustly induced even when translation is inhibited, indicating that nascent/new species cannot be the sole HSR trigger." Please provide reference.

"Meanwhile, evidence has accumulated that nascent polypeptides and newly synthesized proteins, whose folding and assembly may be more easily perturbed during stress (20)". Please include additional references showing that nascent polypeptides challenge protein homeostasis upon stress and are compartmentalized at specific condensates, such as for example PMID: 31271238.

"Finally, many of the proteins which condense in response to stress are translation initiation, elongation, or ribosome biogenesis factors, whose condensation—and likely inactivation—accompanies suppression of the associated processes." Please provide reference such as for example PMID: 30082464.

"Hsf1 targets include many molecular chaperones, including Hsp70, known to regulate condensation". This sentence may be misinterpreted. Both in yeast and mammalian cells, HSP70 has been shown to facilitate the disassembly of specific types of condensates, including stress granules; experimental evidence has been published by the authors (PMID: 35148816), as well as by other independent groups (PMID: 27570075); yet, HSP70 is not required for their formation (condensation). Please rephrase and provide additional references.

"Hsp70 is also capable of playing the essential role of Hsp70 in dispersing heat-triggered *S. cerevisiae* Pab1 condensates". Please verify this sentence.

Reviewer #3 (Remarks to the Author):

The article explores the use of HX-MS to compare structural differences in protein condensates due to temperature stress. The data obtained are interesting.

I recommend the inclusion of a supplementary figure showing the peptide coverage map following HX data curation for each species of Pab1. This would be appropriate in order to align with the community standards as explained in the following article. Masson G.R., et al. Recommendations for performing, interpreting and reporting hydrogen deuterium exchange mass spectrometry (HDX-MS) experiments. *Nature Methods* 2019, 16 (7), 595-602.

Consider using yellow/blue scheme in figure 5 to indicate difference in %D for three Pab1 orthologs (as done in many HDX papers). The selected color scheme may not be accessible to individuals with reduced color perception. (Wong, B., Author Correction: Points of view: Color blindness. *Nature Methods* 2023, 20 (8), 1266-1266.)

REVIEWER COMMENTS

We thank the reviewers for their thoughtful comments, and have additionally edited the manuscript to address each of their concerns, reviewed point-by-point below.

Reviewer #1

Comment: In the manuscript “An adaptive biomolecular condensation response is conserved across environmentally divergent species,” Kik et al investigate the evolutionary conservation of the stress-triggered condensation proteome. They focus on comparing responses in related *Saccharomyces* species chosen to represent different thermal niches and also genome duplication status. They compare the temperature-dependent changes in the transcriptomes and condensed-phase proteomes to assess global changes in cellular composition in response to heat shock. They first establish that the three species show heat shock responses at different temperatures that in each case are few degrees higher than the temperature at which they grow maximally. They observe that Hsp70 response and changes in Hsf1 regulated genes show similar response patterns albeit at the corresponding different heat shock temperatures for the three species. While the pre- whole genome duplication species *K. marxianus* deviates from the transcript abundance trends for *Msn2/4* target orthologs upon heat shock seen in the two other species, the authors attribute this difference to its respiratory lifestyle. Except for these targets, targets of other transcription factors appear to show an evolutionarily-conserved transcriptional response.

The authors then use LC-MS/MS of fractionated lysates upon heat shock to evaluate changes in solubility of proteins. They report that many orthologous proteins known to show heat-induced condensation, in particular those that have been described as super-aggregators lose solubility upon heat shock and do so at the species-specific heat shock temperatures. In contrast glycolytic proteins remain soluble suggesting this is not a general denaturation and aggregation phenomenon. Next the authors compare the characteristics of Pab1 condensation across the test species building off the models of Pab1 function the group has described in *S. cerevisiae*. They examine the impact of hydrophobic residues on the temperature sensitivity of Pab1 by making mutating them to alter their hydrophobicity. They report that all changes that decrease hydrophobicity similarly decrease the temperature threshold for condensation. They see the opposite trend with increased hydrophobicity. Reduced hydrophobicity also impacted fitness under heat shock, suggesting that condensation of Pab1 promotes heat shock tolerance in *S. cerevisiae* and *S. kudriavzevii*. Finally they investigate the structural basis of Pab1 condensation using deuterium exchange MS to examine the structural dynamics of different regions of Pab1. The authors conclude that stress-triggered condensation is an evolutionary conserved adaptive sensing program upstream of transcriptional response.

Response: We thank the reviewer for their close reading of the manuscript, and find this summary captures the essence of our study.

Comment: The work is thorough and presented clearly. Although it doesn't add substantially to a mechanistic understanding of the stress-response, it shows evolutionary conservation of the mechanism. While I wouldn't require this in a revision, I am curious about the recovery from stress and if the reverse steps are comparable conserved. Are the super aggregators reversible upon return to permissive temperature? It would be interesting to perform the proteomics on the soluble and insoluble fractions following a recovery. In general, this is a well done, well written set of experiments that is an important example of how evolutionarily divergent organisms can be a useful lens to understand the process of stress adaptation and biomolecular condensation. This approach is especially important as the field grapples with decoding the molecular grammar of IDRs.

Response: We agree with the reviewer's assessment of the novelty of the work in describing the evolutionary conservation of the stress response. In terms of the post-stress recovery behavior of condensates, previous work from our group has shown that, in *S. cerevisiae*, indeed superaggregators as well as other stress-sensitive proteins are returned to solubility post-stress (see (Wallace et al. 2015), Figure 6). In this way, the process is reversible. While we are only aware of this type of data for *S. cerevisiae*, given those results, we would hypothesize that we will observe similar recovery behavior in the other two species. We appreciate the reviewers comment and agree that is an additional angle for future work.

Reviewer #2

Comment: In this manuscript, Samantha Keyport Kik and colleagues used three evolutionarily diverged budding yeast species adapted to different temperatures (cryophilic, mesophilic, and thermotolerant) to understand whether protein condensation may act as a sensor of and coordinate the stress responses at the transcriptional and translational levels. The manuscript is well-thought, well-organized and addresses an important question.

Response: We thank the reviewer for their positive comments about the manuscript.

Comment: There are two aspects that may deserve more detailed investigation and discussion. First, using purified Pab1 from the three different species, which show 69-98% sequence identity, the authors could show that temperature sensitivity and condensation at a specific temperature is encoded in the primary protein sequence, similar to what already published for Ded1.

Response: Indeed, the Ded1 story (Iserman et al. 2020) provided strong motivation and guidance for the way we approached this work. The *in vitro* work on purified Pab1, as shown in Figures 4a-d, and especially Figure 4a, is sufficient evidence that temperature sensitivity is encoded in primary sequence. In the revised manuscript we write:

We conclude that Pab1's temperature sensitivity and tuning to a thermal niche are both largely encoded directly in its amino-acid sequence, paralleling results from the RNA helicase Ded1 (Iserman et al. 2020).

Comment: The authors identify a group of proteins, called “superaggregators”, that condense and lose solubility upon heat shock. Do superaggregators include proteins that play a primary role in the process of condensation and proteins that act as clients and are recruited inside forming condensates (proteins that co-condense)? Or do they exclude proteins known to co-condense (without triggering by themselves the formation of condensates)? It would be informative, based on the existing literature, to stress the difference between these two types of proteins (condensing and co-condensing), in order to verify whether the optimal temperature required for condensation does depend on the exact sequence also for the co-condensing proteins or only for the condensing ones.

Response: We thank the reviewer for their attention to detail on this topic. It is true that autonomous condensation (i.e., inherent heat sensitivity) and co-condensation (i.e., merely associations with an autonomous condenser) are mechanistically distinct: in one case, the condensation is sequence encoded, and in the other, protein is merely associated with an autonomous condenser and thus is resolved as a superaggregator. In short, our biochemical fractionation method is unable to differentiate these phenomena. In this sense, all proteins in the insoluble, pellet fraction who meet the superaggregator definition set forth in (Wallace et al. 2015) will be considered as such, including autonomous condensers and their passengers. We have reasoned the way to differentiate such proteins involves *in vitro* methods, such as DLS as we have executed for Pab1. We have added the following language in the text to clarify this important point:

Many purified proteins identified using this method have been shown to condense in vitro when exposed to physiological heat shock—including poly(A)-binding protein Pab1 and aminoacyl tRNA synthetase complex components Gus1, Arc1, and Mes1 (Wallace et al. 2015); RNA helicase Ded1 (Iserman et al. 2020); and poly(U)-binding protein Pub1 (Kroschwald et al. 2018)—with no known exceptions to date. Our MS method cannot distinguish which proteins cause condensation (drivers) rather than bind to such proteins (passengers). In the case of the Gus1/Arc1/Mes1 complex, Gus1 and Mes1 are drivers, whereas Arc1 is a passenger (7), yet all three condense. The question of which proteins form condensates is separate from the question of how condensation occurs.

Comment: Second, by comparing structural differences in condensates relative to monomers using HDX-MS, temperature-dependent condensation and growth rate the authors conclude that condensation in vivo is required for an organism to grow at temperatures above that of optimal growth. Under certain experimental conditions, growth was not resumed during the stress recovery phase, likely due to death (Figure 4). Does death correlate with the formation of irreversible condensates? And does cell growth recovery occur when condensates dissolve during the stress recovery phase? It will be important to address this point, given the accumulating evidence showing that cells ability to respond and adapt to acute environmental stress, including heat shock and acidosis, depends on the formation of reversible amyloidogenic condensates (that are dispersed with the assistance of the protein quality control system) (PMID: 27720612; PMID: 34704593; PMID: 31271238).

Response: We thank the reviewer for this thoughtful and topical set of questions. In terms of the death phenotype, we assume the reviewer is referring to Figure 4e, in the bottom right hand corner where *S. kudriavzevii* exposed to 40 °C for four days leads to inability to resume growth at 23 °C. While it is an interesting idea that this phenotype is due to the formation of irreversible condensates, we would not want to imply this, since any number of other unrelated factors could also contribute; this strain is not able to grow at 40 °C at all in our hands. The revision now has the following addition:

“Growth of S. kudriavzevii does not resume when cells are moved to 23°C after stress at 40°C (Figure 4e). Why these cells cannot resume growth, and whether this behavior has any link to condensation, is unclear; S. kudriavzevii shows no growth at 40°C (Figure 1b), so after four days, the small number of initially deposited cells may simply have become inviable.”

Comment: Figure 4: Dynamic light scattering (DLS) and the increases in Rh have been used here to identify the onset of condensation, as previously established by the authors (Riback et al., Cell 2017). The authors should consider adding representative images showing the Pab1 condensates (WT and mutants) at the different temperatures and in the different species.

Response: We thank the reviewer for the suggestion to add images of the condensates from different species to the manuscript. We have carried out the suggested imaging as a function of temperature for all three species, which produced the expected results and are now reported in Figure S4, along with new methods. The main text has been revised as follows:

DLS is highly sensitive to small condensates of only a few nanometers in size. We thus also looked for the appearance of microscopically visible condensates, which also emerged at temperatures correlated with each species’ thermal niche (Figure S4). Condensates in all species also had similar morphologies, branched chains of small clusters, as previously reported (Riback et al. 2017) (Figure S4).

Comment: "Nevertheless, the HSR can be robustly induced even when translation is inhibited, indicating that nascent/new species cannot be the sole HSR trigger." Please provide reference.

Response: We have added two new references: (Baler, Welch, and Voellmy 1992; Triandafillou et al. 2020).

Comment: "Meanwhile, evidence has accumulated that nascent polypeptides and newly synthesized proteins, whose folding and assembly may be more easily perturbed during stress (20)". Please include additional references showing that nascent polypeptides challenge protein homeostasis upon stress and are compartmentalized at specific condensates, such as for example PMID: 31271238.

Response: We have added several new references to support this claim: (Xu et al. 2016; Mediani et al. 2019; Tye and Churchman 2021; Ali et al. 2023).

Comment: "Finally, many of the proteins which condense in response to stress are translation initiation, elongation, or ribosome biogenesis factors, whose condensation—and likely inactivation—accompanies suppression of the associated processes." Please provide reference such as for example PMID: 30082464.

Response: We have added two references in the text: (Cherkasov et al. 2013; Wallace et al. 2015; Ivanov, Kedersha, and Anderson 2019).

Comment: "Hsf1 targets include many molecular chaperones, including Hsp70, known to regulate condensation". This sentence may be misinterpreted. Both in yeast and mammalian cells, HSP70 has been shown to facilitate the disassembly of specific types of condensates, including stress granules; experimental evidence has been published by the authors (PMID: 35148816), as well as by other independent groups (PMID: 27570075); yet, HSP70 is not required for their formation (condensation). Please rephrase and provide additional references.

Response: Thank you to the reviewer for pointing out this potential misinterpretation. We have rephrased the statement and provided several references: "Moreover, Hsf1 targets include many molecular chaperones, including Hsp70, known to regulate the dispersal of biomolecular condensates (Cherkasov et al. 2013; Ganassi et al. 2016; Yoo et al. 2022)."

Comment: "Hsp70 is also capable of playing the essential role of Hsp70 in dispersing heat-triggered *S. cerevisiae* Pab1 condensates". Please verify this sentence.

Response: We thank the reviewer for catching our error. We have rewritten the sentence to read: "Hsp70 also plays an essential role in dispersing heat-triggered *S. cerevisiae* Pab1 condensates."

Reviewer #3

Comment: The article explores the use of HX-MS to compare structural differences in protein condensates due to temperature stress. The data obtained are interesting.

Response: We thank the reviewer for this perspective on the data.

Comment: I recommend the inclusion of a supplementary figure showing the peptide coverage map following HX data curation for each species of Pab1. This would be appropriate in order to align with the community standards as explained in the following article. Masson G.R., et al. Recommendations for performing, interpreting and reporting hydrogen deuterium exchange mass spectrometry (HDX-MS) experiments. *Nature Methods* 2019, 16 (7), 595-602.

Response: We appreciate this suggestion and have added a supplemental figure (Figure S5) showing the peptide maps for each species.

Comment: Consider using yellow/blue scheme in figure 5 to indicate difference in %D for three Pab1 orthologs (as done in many HDX papers). The selected color scheme may not be accessible to individuals with reduced color perception. (Wong, B., Author Correction: Points of view: Color blindness. *Nature Methods* 2023, 20 (8), 1266-1266.)

Response: Indeed, we have attempted to address this accessibility issue by using colors from the viridis palette, which was created specifically to deal with this unintended issue ([doi:10.5281/zenodo.4679423](https://doi.org/10.5281/zenodo.4679423), <https://cran.r-project.org/web/packages/viridis/vignettes/intro-to-viridis.html>), throughout the manuscript. All of the figures in the paper, including Figure 5, should be accessible to individuals with reduced color perception.

References Cited

- Ali, Asif, Rania Garde, Olivia C. Schaffer, Jared A. M. Bard, Kabir Husain, Samantha Keyport Kik, Kathleen A. Davis, et al. 2023. "Adaptive Preservation of Orphan Ribosomal Proteins in Chaperone-Dispersed Condensates." *Nature Cell Biology*, October, 1–13.
- Baler, R., W. J. Welch, and R. Voellmy. 1992. "Heat Shock Gene Regulation by Nascent Polypeptides and Denatured Proteins: hsp70 as a Potential Autoregulatory Factor." *The Journal of Cell Biology* 117 (6): 1151–59.
- Cherkasov, Valeria, Sarah Hofmann, Silke Druffel-Augustin, Axel Mogk, Jens Tyedmers, Georg Stoecklin, and Bernd Bukau. 2013. "Coordination of Translational Control and Protein Homeostasis during Severe Heat Stress." *Current Biology: CB* 23 (24): 2452–62.
- Ganassi, Massimo, Daniel Mateju, Ilaria Bigi, Laura Mediani, Ina Poser, Hyun O. Lee, Samuel J. Seguin, et al. 2016. "A Surveillance Function of the HSPB8-BAG3-HSP70 Chaperone Complex Ensures Stress Granule Integrity and Dynamism." *Molecular Cell* 63 (5):

796–810.

- Iserman, Christiane, Christine Desroches Altamirano, Ceciel Jegers, Ulrike Friedrich, Taraneh Zarin, Anatol W. Fritsch, Matthäus Mittasch, et al. 2020. “Condensation of Ded1p Promotes a Translational Switch from Housekeeping to Stress Protein Production.” *Cell* 0 (0). <https://doi.org/10.1016/j.cell.2020.04.009>.
- Ivanov, Pavel, Nancy Kedersha, and Paul Anderson. 2019. “Stress Granules and Processing Bodies in Translational Control.” *Cold Spring Harbor Perspectives in Biology* 11 (5). <https://doi.org/10.1101/cshperspect.a032813>.
- Kroschwald, Sonja, Matthias C. Munder, Shovamayee Maharana, Titus M. Franzmann, Doris Richter, Martine Ruer, Anthony A. Hyman, and Simon Alberti. 2018. “Different Material States of Pub1 Condensates Define Distinct Modes of Stress Adaptation and Recovery.” *Cell Reports* 23 (11): 3327–39.
- Mediani, Laura, Jordina Guillén-Boixet, Jonathan Vinet, Titus M. Franzmann, Ilaria Bigi, Daniel Mateju, Arianna D. Carrà, et al. 2019. “Defective Ribosomal Products Challenge Nuclear Function by Impairing Nuclear Condensate Dynamics and Immobilizing Ubiquitin.” *The EMBO Journal* 38 (15): e101341.
- Riback, Joshua A., Christopher D. Katanski, Jamie L. Kear-Scott, Evgeny V. Pilipenko, Alexandra E. Rojek, Tobin R. Sosnick, and D. Allan Drummond. 2017. “Stress-Triggered Phase Separation Is an Adaptive, Evolutionarily Tuned Response.” *Cell* 168 (6): 1028–40.e19.
- Triandafillou, Catherine G., Christopher D. Katanski, Aaron R. Dinner, and D. Allan Drummond. 2020. “Transient Intracellular Acidification Regulates the Core Transcriptional Heat Shock Response.” *eLife* 9 (August). <https://doi.org/10.7554/eLife.54880>.
- Tye, Blake W., and L. Stirling Churchman. 2021. “Hsf1 Activation by Proteotoxic Stress Requires Concurrent Protein Synthesis.” *Molecular Biology of the Cell* 32 (19): 1800–1806.
- Wallace, Edward W. J., Jamie L. Kear-Scott, Evgeny V. Pilipenko, Michael H. Schwartz, Pawel R. Laskowski, Alexandra E. Rojek, Christopher D. Katanski, et al. 2015. “Reversible, Specific, Active Aggregates of Endogenous Proteins Assemble upon Heat Stress.” *Cell* 162 (6): 1286–98.
- Xu, Guilian, Amrutha Pattamatta, Ryan Hildago, Michael C. Pace, Hilda Brown, and David R. Borchelt. 2016. “Vulnerability of Newly Synthesized Proteins to Proteostasis Stress.” *Journal of Cell Science* 129 (9): 1892–1901.
- Yoo, Haneul, Jared A. M. Bard, Evgeny V. Pilipenko, and D. Allan Drummond. 2022. “Chaperones Directly and Efficiently Disperse Stress-Triggered Biomolecular Condensates.” *Molecular Cell* 82 (4): 741–55.e11.

REVIEWERS' COMMENTS

Reviewer #2 (Remarks to the Author):

The authors have adequately addressed all my concerns.

Reviewer #3 (Remarks to the Author):

The suggestions have been addressed by the authors. Thank you for taking the time for this.